# VAGEN: Reinforcing World Model Reasoning for Multi-Turn VLM Agents

**Kangrui Wang**[1*] **Pingyue Zhang**[1*] **Zihan Wang**[1*] **Yaning Gao**[5*] **Linjie Li**[2*] **Qineng Wang**[1]
**Hanyang Chen**[6] **Chi Wan**[1] **Yiping Lu**[1] **Zhengyuan Yang**[4] **Lijuan Wang**[4]
**Ranjay Krishna**[2] **Jiajun Wu**[3] **Li Fei-Fei**[3] **Yejin Choi**[3] **Manling Li**[1†]

[1]Northwestern University   [2]University of Washington   [3]Stanford University   [4]Microsoft
[5]University of Wisconsin-Madison   [6]University of Illinois Urbana-Champaign

http://mll.lab.northwestern.edu/VAGEN
kangrui.wang@northwestern.edu, manling.li@northwestern.edu

## Abstract

A major challenge in training VLM agents, compared to LLM agents, is that states shift from simple texts to complex visual observations, which introduces partial observability and demands robust world modeling. We ask: *can VLM agents build internal world models through explicit visual state reasoning?* In this work, we architecturally enforce and reward VLM agent's reasoning process via reinforcement learning (RL), formulating the problem as a Partially Observable Markov Decision Process (POMDP). We demonstrate that structuring agent's reasoning into `StateEstimation` ("what is the current state?") and `TransitionModeling` ("what is next?") is critical by studying five reasoning strategies. Investigating how agents should ground visual states and represent these internal beliefs, we reveal the optimal representations are task-dependent: *Natural Language* excels at capturing semantic relationships for general tasks, while *Structured* formats are essential for high-precision manipulation. These insights motivate our approach to reward shaping and credit assignment. We leverage a `WorldModeling` Reward to densely rewards the agent's turn-by-turn state predictions, while our Bi-Level General Advantage Estimation (Bi-Level GAE) enables turn-aware credit assignment. Through such world model reasoning, we enable a 3B model to achieve performance of 0.82 on a set of five diverse agent tasks, nearly $3\times$ improvement over its untrained counterpart (0.21) and surpassing proprietary reasoning models like GPT-5 (0.75), Gemini 2.5 Pro (0.67) and Claude 4.5 (0.62). All experiments are supported by our VAGEN framework, a scalable system for training and analyzing multi-turn VLM agents across diverse visual environments.

## 1 Introduction

The core challenge in multi-turn agentic tasks is accurate interpretation and tracking dynamic environments, which becomes significantly harder when the agents sense the world through vision rather than text. Vision-language Model (VLM) agentic tasks are inherently complex due to the challenges in understanding visual states, which often are partial and noisy *Observations*, fundamentally reframing the problem from an Markov Decision Process (MDP) to a more challenging Partially Observable Markov Decision Process (POMDP). Within a POMDP, the agent is no longer simply acting, but first estimating the true state of the world from observations. The bridge connecting *what an agent sees* to *what it needs to know* is the central focus of our work: the internal World Model [1]. It functions as a cognitive engine, maintaining an *internal belief* over the true state based on visual observations. While VLMs agents have been proposed for visual agentic such as like games [2–4], embodied

---

[*]Equal contribution.
[†]Corresponding author.

39th Conference on Neural Information Processing Systems (NeurIPS 2025).

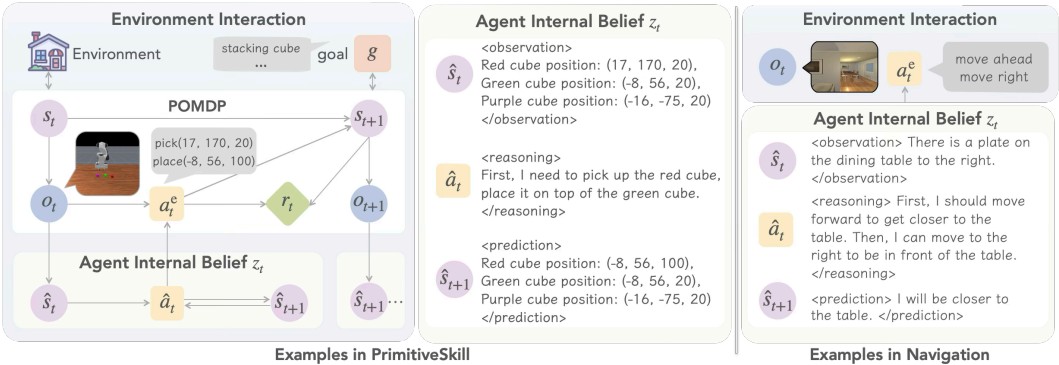

**Figure 1:** Overview of the VAGEN Framework. Within a POMDP, VLM Agents build internal world models through explicit visual state reasoning over the `StateEstimation` $P(\hat{s}_t|o_t)$ and `TransitionModeling` $P(\hat{s}_{t+1}|o_t, \hat{s}_t, \hat{a}_t)$, where $\hat{s}_t$ denotes the internal belief of the state, $\hat{a}_t$ represents the internal belief of the action, and $a_t^e$ is the parsed executable action, as detailed in Table 4.

AI [5–7], and computer use [8–10], current approaches in such multi-turn agentic tasks often lack explicit internal world modeling to strengthen visual state reasoning. This lead us to think: *How can we effectively teach VLMs to build internal world models through explicit visual state reasoning?*

To address this, we reward VLMs via reinforcement learning (RL) to encourage their thinking process to be structured into `StateEstimation` (describing the current visual state) and `TransitionModeling` (predicting next state), which are the essential two components of `WorldModeling` [1]. We focus on multi-turn RL to optimize the entire interaction trajectory [11–16, 14, 17], so that the agent can maintain and update internal beliefs as interactions unfold. To validate this principled reasoning structure, we systematically compare five reasoning strategies via controlling format reward in RL: `NoThink` [18], `FreeThink` [19, 20] as implicit visual state reasoning, and explicit reasoning through `StateEstimation` via `<observation>` tokens, `TransitionModeling` via `<prediction>` tokens, and their combination, `WorldModeling`. Our findings indicate that incorporating explicit visual state reasoning like `StateEstimation` and `TransitionModeling` into VLM's thinking process during RL training can enhance task performance. Notably, the full reasoning strategy, `WorldModeling`, achieves an overall performance of $0.76$, yielding better results than `FreeThink` ($0.67$) while clearly surpassing `NoThink` ($0.28$).

Building on the insight that VLM agents might make better decisions when reasoning about what they see now (`StateEstimation`) and what they will see next (`TransitionModeling`), a foundation question lies in representation: for an agent to "think" about the visual world, *what is the optimal representation for its internal monologue?* To translate pixels into a mental model it can reason upon, we explore a spectrum of representation possibilities and our findings point to a crucial design principle that the choice of representation is not universal but is dictated by the task's demands. **Natural language** excels in a general-purpose tasks by providing the robust and inherent semantic-centric representation, allowing the VLM to use the vast pre-trained knowledge. However, its inherent ambiguity make it unsuitable for high-precision tasks such as robot manipulation, where **structured formats** providing exact coordinates is essential. As a third alternative, **symbolic representation** offers a more abstract representation. Surprisingly, this abstraction does not produce a more generalizable solution. Instead, it proves to be the least effective method in our experiments, creating a grounding problem where the models struggle to connect abstract symbols to raw visual input without targeted training.

Having established what VLM agents should reason about and in what ways they can reason, the remaining question is how to effectively optimize such reasoning via reward shaping and advantage estimation. To this end, we introduce a turn-level `WorldModeling` Reward, a dense reward derived from an LLM-as-a-Judge [21] framework, which evaluates the accuracy of the agent's explicit state descriptions and predictions against ground-truth states. Further, credit assignment in multi-turn settings can be optimized through our proposed Bi-Level General Advantage Estimation (Bi-Level GAE) via encouraging credit propagation in each single-step world modeling. Standard GAE [22] methods, which compute advantages token-by-token in a backward manner from the end of a trajectory, would lead to unstable reward propagation because the sparse end-of-trajectory signal must travel across long horizons. To address this, we first compute advantages at the turn level to assess whether a VLM's response in a single-step world modeling is generally effective, and then propagate

these signals to the token level, providing each VLM-generated token with fine-grained advantages to optimize its generation process. Our results demonstrate that our VAGEN-Full approach, with `WorldModeling` Reward and Bi-Level GAE, consistently outperforms the VAGEN-Base without these mechanisms, leading to improved reasoning quality and higher task success rates by providing the dense reward with the adapted delivery mechanism.

We present VAGEN, a scalable training framework that decouples environment setup from model training, enabling efficient experimentation and algorithmic extensibility. Together, this work establish a principled pathway for developing VLM agents capable of building intern world models through explicit visual reasoning.

## 2 Build Internal World Models via Visual State Reasoning in Multi-Turn RL

### 2.1 Problem Formulation

We frame the multi-turn VLM agentic tasks as a Partially Observable Markov Decision Process (POMDP) [23], represented by the tuple $(\mathcal{S}, \mathcal{O}, \mathcal{A}, P, R, \Omega, \gamma)$. Here, $\mathcal{S}$ denotes the environment state space, $\mathcal{O}$ is the space of observations perceived by the agent, and $\mathcal{A}$ is the space of actions. At each turn $t$, the agent produces an action $a_t \in \mathcal{A}$. In response, the environment transitions from state $s_t$ to a new state $s_{t+1}$ according to the state transition function $P(s_{t+1}|s_t, a_t)$ and emits a scalar reward $r_t = R(s_t, a_t)$. The agent then receives a new observation $o_{t+1} \in \mathcal{O}$, which is a partial view of the new state, sampled from $\Omega(\cdot|s_{t+1})$. The agent's objective is to learn a policy $\pi_\theta$ that maximizes the expected cumulative discounted return over a trajectory, $\max_\theta \mathbb{E}_{\pi_\theta, P, \Omega} \left[ \sum_t \gamma^t r_t \right]$, where $\gamma \in [0, 1]$ is the discount factor. In the VLM agent setting, the policy $\pi_\theta$ is parameterized by a VLM that takes in images and textual descriptions as observations, and outputs language token sequences as actions. A summary of important notations can be found in Table 4. A detailed discussion of our POMDP formulation and its motivation can be found in Appendix A.1.

### 2.2 Visual State Reasoning as an Internal World Model

Existing RL frameworks for VLM reasoning [24] are primarily based on single-turn optimization, which limits their ability to capture evolving interaction context. To better address the demands of multi-turn agentic tasks, we optimize the entire interaction trajectory, inspired by [11, 25]. We propose to maintain an *internal belief* of its interaction that evolves over time, and we design a training framework that integrates *world modeling* into multi-turn trajectory optimization for visual state reasoning, as shown in Figure 1. Our implementation is based on verl[26].

**Reasoning Trajectory Rollout for Multi-Turn VLM Agents.** Each trajectory begins with an initial state $s_0$, observation $o_0$ and goal $g$. The observation $o_t$ is visual images observed at turn $t$ for VLM agents, which we normally input to VLMs together with corresponding textual prompts. The VLM agent then generates an action $a_t$ using the current policy $\pi_\theta$. To reason about visual states and build internal beliefs, the generated action $a_t$ is a sequence of text tokens including both reasoning tokens $z_t$ and executable action $a_t^e$:

$$a_t = \langle z_t, a_t^e \rangle.$$

After $a_t^e$ is parsed and executed, the environment produces a reward $r_t$ as the feedback and transitions to a new state $s_{t+1}$, providing a new observation $o_{t+1}$ to the agent. This process is repeated over $T$ turns to collect a trajectory $\tau = (s_0, o_0, a_0, r_0, \ldots, s_{T-1}, o_{T-1}, a_{T-1}, r_{T-1}, s_T, o_T)$.

**Visual State Reasoning as World Modeling.** Instead of just acting to what the VLM agent sees, we explicitly reason about visual states to build an *internal belief* of the environment. In agentic tasks, especially those defined as POMDPs, the agent's visual observation $o_t$ offers a view of the true world state $s_t$. To act effectively, the agent should interpret what the current visual observation actually represents and predict future state changes. We train the VLM agent to do this by explicitly generating its reasoning tokens $z_t$ as building up an internal belief of a structured world model [1], including a state model that interprets the present (`StateEstimation`) and a transition model that predicts the future (`TransitionModeling`). In detail, `StateEstimation` grounds visual observations $o_t$ into a state belief $\hat{s}_t$ that approximates the hidden true state $s_t$, highlighting the central challenge of POMDPs compared to fully observable MDPs. Meanwhile, `TransitionModeling` reasons over the belief of the potential next state $\hat{s}_{t+1}$, allowing the agent to simulate internally how its action at turn $t$ will transition the states. This predictive step is crucial for planning of a multi-turn trajectory. We implement multiple reasoning strategies, ranging from minimal (`NoThink`, `FreeThink`) to structured

world modeling (`StateEstimation`, `TransitionModeling`, `WorldModeling`), by shaping the reasoning tokens $z_t$ differently to capture and update internal beliefs about visual states.

1. `NoThink`: we train the VLM agent to generate only an executable action $a_t^e$, and the output action token $a_t$ is `<think></think><answer>`$a_t^e$`</answer>`,

$$z_t = \emptyset$$

2. `FreeThink`: we train the VLM agent to produce any form of natural language reasoning, allowing visual state reasoning to emerge without a predefined structure. The agent generates action tokens as `<think>`$z_t$`</think> <answer>`$a_t^e$`</answer>`,

$$z_t \neq \emptyset, \text{ and } z_t \text{ is natural language tokens.}$$

3. `StateEstimation`: we train the VLM agent to explicitly verbalize the current state belief $\hat{s}_t$ given the visual observation $o_t$, to approximate the true underlying state $s_t$ and help better predict the executable action $a_t^e$. The action token $a_t$ is formatted as `<think><observation>` $\hat{s}_t$ `</observation> <reasoning>` $\hat{a}_t$ `</reasoning> </think> <answer>`$a_t^e$`</answer>`,

$$z_t = \langle \hat{s}_t, \hat{a}_t \rangle, \text{ learning to approximate } \hat{s}_t \to s_t$$

4. `TransitionModeling`: we train the VLM agent to explicitly simulate in its internal belief space about the next state $\hat{s}_{t+1}$, which helps reason over the executable action $a_t^e$ with best expected reward. The output action token $a_t$ is structured as `<think><reasoning>` $\hat{a}_t$ `</reasoning>` `<prediction>`$\hat{s}_{t+1}$ `</prediction> </think> <answer>`$a_t^e$`</answer>`, [3]

$$z_t = \langle \hat{a}_t, \hat{s}_{t+1} \rangle, \text{ learning to approximate } \hat{s}_{t+1} \to s_{t+1}$$

5. `WorldModeling`: $z_t$ is required to both describe the current state and predict the next state: `<think><observation>`$\hat{s}_t$ `</observation> <reasoning>` $\hat{a}_t$ `</reasoning>` `<prediction>` $\hat{s}_{t+1}$ `</prediction> </think> <answer>`$a_t^e$`</answer>`,

$$z_t = \langle \hat{s}_t, \hat{a}_t, \hat{s}_{t+1} \rangle, \text{ learning to approximate } \hat{s}_t \to s_t, \hat{s}_{t+1} \to s_{t+1}$$

To encourage strict adherence to such a structured action format, we incorporate a format reward $r_t^{\texttt{format}}$ during training, following the strategy of DeepSeek-R1 [20]. The reward design is detailed in Section 2.3.

**Policy Optimization.** Once trajectories are collected, we begin the optimization phase using an actor-critic approach. The actor's policy $\pi_\theta$ is updated using the Proximal Policy Optimization (PPO) objective [27]. Denote $\bar{\tau}$ as a token sequence converted from the trajectory $\tau$ with an encoder $\mathcal{E}$, $u_i(\theta) = \frac{\pi_\theta(\bar{\tau}_i | \bar{\tau}_{<i})}{\pi_{\text{old}}(\bar{\tau}_i | \bar{\tau}_{<i})}$ as the probability ratio between the current and old policies, and let $\bar{\tau}_{<i}$ denotes the prefix of token $i$. The PPO loss is defined as:

$$J^{\text{PPO}}(\theta) = \frac{1}{\sum_i M_i^{\text{loss}}} \sum_i M_i^{\text{loss}} \cdot \min\left(u_i(\theta) A_i, \text{clip}(u_i(\theta), 1 - \varepsilon, 1 + \varepsilon) A_i\right),$$

where $M_i^{\text{loss}}$ is a mask that is 1 for action tokens and 0 for observation tokens, $A_i$ is the per-token advantage and $\varepsilon$ is a clipping hyperparameter.

**Advantage and Value Estimation.** Concurrently, the critic parameters $\phi$ of the value function $V_\phi$ are updated by minimizing the squared error between its predictions and the target values $Y_i$:

$$J^{\text{Critic}}(\phi) = \frac{1}{\sum_i M_i^{\text{loss}}} \sum_i M_i^{\text{loss}} \cdot \left(V_\phi(\bar{\tau}_{<i+1}) - Y_i\right)^2$$

The actor and critic updates require computing per-token advantages $A_i$ and target values $Y_i$. In our VAGEN-Base setting, we use Token-Level Generalized Advantage Estimation (GAE) [22]. At each token index $i$, we apply a per-token KL penalty that encourages the current policy $\pi_\theta$ to stay close to a frozen reference policy $\pi_{\text{ref}}$. The penalty is scaled by a coefficient $\beta > 0$. For all intermediate

---

[3]For simplicity, we represent the belief of next state in $t$-th turn as $\hat{s}_{t+1}$. Please note that in the turn of $t + 1$, we re-generate state belief $\hat{s}_{t+1}$ given the parallel observation $o_{t+1}$, rather than using the predicted state belief in the $t$-th turn.

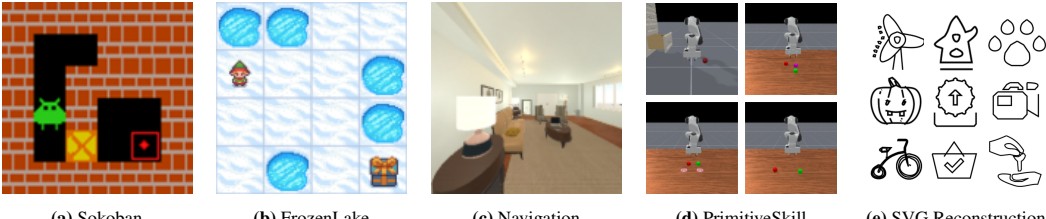

|  **(a)** Sokoban | **(b)** FrozenLake | **(c)** Navigation | **(d)** PrimitiveSkill | **(e)** SVG Reconstruction |

**Figure 2:** Examples of visual states from five environments used in our study.

action tokens, reward $r_i$ is set to the KL penalty. At the final action token $I$, reward $r_I$ is set as the sum of KL penalty and total trajectory return $R(\tau) = \sum_{t=0}^{T-1} R(s_t, a_t)$.

We then calculate the temporal-difference (TD) error using a discount factor $\gamma$ and the GAE parameter $\lambda$:

$$\delta_i = r_i + \gamma V_\phi(\bar{\tau}_{<j}) - V_\phi(\bar{\tau}_{<i})$$

$$A_i = \delta_i + \gamma\lambda A_j$$

This recursion is initialized at the end of the sequence with $A_I = \delta_I$, $j$ denotes the index of the next action token after token $i$ (skipping the observation tokens). The target value for the critic update is defined as

$$Y_i = A_i + V_\phi(\bar{\tau}_{<i}).$$

The iterative process of trajectory collection, advantage estimation, and policy update continues until convergence. We present a detailed illustration of our multi-turn agent RL framework in Algorithm 1.

## 2.3 Reward Design in Different Environments and Tasks for VLM Agents

To systematically analyze the learning dynamics and visual reasoning capabilities of VLM agents, we developed an evaluation suite featuring five distinct agentic tasks (Figure 2). These tasks were chosen to cover a wide range of challenges, including diverse visual state representations and action spaces: **Classic Grid Puzzles** (Sokoban and FrozenLake), **Embodied 3D Navigation** (Navigation), **Detailed Object Manipulation** (PrimitiveSkill), and **Abstract Geometric Reconstruction** (SVG Reconstruction).

**Sokoban** [28]: In this classic puzzle, the agent must push all boxes to target locations. The visual state is a 2D grid, and the action space is discrete (`up`, `down`, `left`, `right`).

**FrozenLake** [29]: The agent navigates a 2D grid to reach a goal while avoiding holes. The visual state and discrete action space are similar to Sokoban. We disable the "slippery" setting for determinism.

**Navigation** [7, 30]: In this 3D embodied task, the agent follows instructions to find an object, perceiving the world through a first-person view and using discrete actions (e.g., `moveahead`).

**PrimitiveSkill** [31–34]: The agent controls a Panda Arm to perform complex manipulation, using a hybrid action space (e.g., `pick(x,y,z)`). The agent must ground objects in the third-person 3D scene to a list of coordinates.

**SVG Reconstruction** [9]: The agent's goal is to generate SVG code that replicates a target image using an open-ended text action space.

**Reward Strategy and Metrics.** For the SVG task, the reward $R(s_t, a_t)$ is a dense similarity between the generated image $I_{\text{gen}}$ and the target $I_{\text{target}}$, computed as a weighted combination of DreamSim and DINO scores. Task performance is reported as the average DreamSim $\mathbb{E}_{\tau\sim\mathcal{D}}[g(\tau)]$ and DINO $\mathbb{E}_{\tau\sim\mathcal{D}}[h(\tau)]$ similarity of the final output. For the other tasks, $R(s_t, a_t)$ is a scaled binary reward (e.g., $\{0, 10\}$) indicating whether the trajectory completes the objective, and performance is measured by the average success rate $\mathbb{E}_{\tau\sim\mathcal{D}}[f(\tau)]$, where $f(\tau) \in \{0, 1\}$. We modified some environments form original setting to enable efficient RL training, full implementation details are provided in Appendix A.4.

---

[4]Due to Gemini 2.5 Pro's safety policy, a subset of evaluation responses for PrimitiveSkill and Navigation was blocked; reported results are based solely on the unblocked test cases.

**Table 1:** Reasoning on internal world models given visual states, including both `StateEstimation` and `TransitionModeling`, can significantly improve the RL performance. Test success rates are reported for Sokoban, FrozenLak, Navigation and PrimitiveSkill. Test Dino and DreamSim scores are reported for SVG Reconstruction task. Best performance is in **bold**. We gray out results without controlled variables (i.e., same size and architecture), shown for reference only. VAGEN-Full with world modeling enhancement on reward shaping and credit assignment will be introduced in Sec 4 so we also gray it out, shown as a reference.

| Model/Method | Sokoban | FrozenLake | Navigation | | | PrimitiveSkill | | | | | SVG | | | Overall |
|---|---|---|---|---|---|---|---|---|---|---|---|---|---|---|
| | | | Base | Common | **Average** | Place | Stack | Drawer | Align | **Average** | Dino | DreamSim | **Average** | |
| *Open-Source Models* | | | | | | | | | | | | | | |
| Qwen2.5-VL-72B [35] | 0.18 | 0.44 | 0.72 | 0.75 | 0.73 | **1.00** | 0.50 | 0.00 | **1.00** | 0.44 | 0.89 | 0.62 | 0.76 | 0.51 |
| Qwen2.5-VL-7B [35] | 0.13 | 0.14 | 0.28 | 0.39 | 0.34 | 0.00 | 0.00 | 0.00 | 0.75 | 0.19 | 0.83 | 0.28 | 0.55 | 0.27 |
| Qwen2.5-VL-3B [35] | 0.14 | 0.14 | 0.22 | 0.27 | 0.24 | 0.00 | 0.00 | 0.00 | 0.00 | 0.00 | 0.80 | 0.27 | 0.54 | 0.21 |
| VLM-R1-3B [24] | 0.13 | 0.13 | 0.31 | 0.34 | 0.33 | 0.00 | 0.00 | 0.00 | 0.00 | 0.00 | 0.81 | 0.28 | 0.55 | 0.23 |
| *VAGEN: Multi-Turn RL with World Model Reasoning for Visual States (Backbone: Qwen2.5-VL-3B)* | | | | | | | | | | | | | | |
| FreeThink | 0.57 | 0.68 | 0.67 | 0.67 | 0.67 | **1.00** | 0.63 | 0.00 | **1.00** | 0.66 | 0.91 | 0.64 | 0.78 | 0.67 |
| NoThink | 0.57 | 0.09 | 0.00 | 0.00 | 0.00 | 0.00 | 0.00 | 0.00 | 0.00 | 0.00 | 0.89 | 0.62 | 0.76 | 0.28 |
| StateEstimation | 0.56 | 0.68 | 0.78 | 0.69 | 0.74 | 0.00 | 0.00 | 0.00 | 0.00 | 0.00 | 0.92 | 0.64 | 0.78 | 0.56 |
| TransitionModeling | 0.41 | 0.76 | 0.67 | 0.59 | 0.62 | **1.00** | 0.63 | 0.63 | **1.00** | 0.82 | 0.89 | 0.64 | 0.77 | 0.68 |
| WorldModeling | 0.61 | 0.71 | 0.78 | 0.80 | 0.79 | **1.00** | 0.88 | 0.88 | 0.88 | **0.91** | 0.90 | 0.65 | 0.78 | **0.76** |
| VAGEN-Full | 0.79 | 0.74 | 0.80 | 0.81 | 0.81 | 1.00 | 0.88 | 1.00 | 1.00 | 0.97 | 0.91 | 0.67 | 0.79 | 0.82 |
| *RL Baselines with World Model Reasoning Strategy (Backbone: Qwen2.5-VL-3B)* | | | | | | | | | | | | | | |
| Vanilla-PPO | 0.18 | 0.21 | 0.32 | 0.25 | 0.29 | 0.00 | 0.00 | 0.00 | 0.00 | 0.00 | 0.83 | 0.44 | 0.64 | 0.26 |
| GRPO w/ Mask | 0.20 | 0.57 | **0.88** | 0.81 | **0.85** | 0.00 | 0.00 | 0.00 | **1.00** | 0.25 | 0.92 | 0.66 | **0.79** | 0.54 |
| Turn-PPO w/ Mask | 0.38 | 0.70 | 0.78 | **0.84** | 0.81 | 0.00 | 0.00 | 0.00 | **1.00** | 0.25 | 0.89 | 0.64 | 0.77 | 0.55 |
| *Proprietary Models* | | | | | | | | | | | | | | |
| GPT-5 [36] | **0.70** | 0.77 | 0.75 | 0.81 | 0.78 | **1.00** | 0.63 | 0.00 | **1.00** | 0.66 | 0.95 | 0.75 | 0.85 | 0.75 |
| o3 [37] | 0.60 | 0.78 | **0.81** | 0.75 | 0.78 | **1.00** | 0.63 | 0.00 | **1.00** | 0.66 | 0.92 | 0.71 | 0.82 | 0.73 |
| o4-mini [37] | 0.44 | **0.82** | 0.75 | 0.75 | 0.75 | **1.00** | 0.50 | 0.00 | 0.75 | 0.56 | 0.90 | 0.66 | 0.78 | 0.67 |
| GPT-4o [38] | 0.43 | 0.54 | 0.75 | 0.69 | 0.72 | 0.50 | 0.63 | 0.00 | 0.88 | 0.50 | 0.91 | 0.69 | 0.80 | 0.60 |
| Gemini 2.5 Pro[4][39] | 0.58 | 0.78 | 0.63 | 0.63 | 0.63 | 0.63 | 0.63 | 0.00 | 0.75 | 0.50 | 0.93 | 0.78 | 0.86 | 0.67 |
| Gemini 2.0 [40] | 0.28 | 0.61 | 0.50 | 0.63 | 0.56 | 0.75 | 0.13 | 0.00 | 0.25 | 0.28 | 0.93 | 0.74 | 0.84 | 0.51 |
| Claude 4.5 Sonnet [41] | 0.31 | 0.80 | 0.67 | 0.67 | 0.67 | 0.63 | 0.50 | 0.00 | **1.00** | 0.53 | 0.95 | **0.81** | **0.88** | 0.64 |
| Claude 3.7 Sonnet [42] | 0.25 | 0.69 | 0.48 | 0.47 | 0.47 | 0.63 | 0.13 | 0.00 | **1.00** | 0.44 | 0.94 | 0.77 | 0.85 | 0.54 |

## 2.4 What Can We Reason About Visual States?

**Off-the-Shelf VLMs struggle to solve multi-turn agentic tasks.** We benchmark 7 models including proprietary models and open-source models in Table 1. We use different reasoning strategies to prompt models, which are detailed in Appendix B.1. Notably, VLMs that are trained to solve non-agentic tasks, such as VLM-R1, does not show multi-turn agent task advantage. Instead, most models struggle on these tasks, with the best-performing GPT-5 reaching $0.75$ out of $1$ in overall score. Particularly, no model succeeds on the PrimitiveSkill Drawer task. These results indicate a significant capability gap in current VLMs to reason about complex multi-turn visual agentic tasks.

**World Modeling can improve visual reasoning via multi-turn RL.** To examine whether explicit reasoning about visual states improves performance, we train the Qwen2.5-VL-3B model with five reasoning strategies using VAGEN-Base (Section 2.2). As shown in Table 1, `FreeThink` consistently outperforms `NoThink`, particularly in embodied environments like Navigation and PrimitiveSkill, which indicates the importance of explicit reasoning in multi-turn decision-making tasks.

Among different reasoning strategies, `StateEstimation` and `TransitionModeling` show task-specific strengths. `StateEstimation` performs well in Navigation tasks, where understanding current observations is the key. In contrast, `TransitionModeling` achieves strong results in PrimitiveSkill, where predicting future states is crucial for manipulation. However, each strategy alone may lead to reduced performance in tasks where model's prior is less aligned with the task structure or state complexity.

Ono the other hand, **the combined** `WorldModeling` **strategy results in strong and stable performance across all tasks**, as the trained model achieves a substantial improvement over its untrained counterpart ($+0.55$), and even outperforms all proprietary models despite smaller scale. These results demonstrate that explicitly visual states reasoning is crucial for VLM agents. In the following studies, we use `WorldModeling` as the general visual reasoning strategy.

**Table 2:** Examples of how visual states are converted into internal beliefs using natural language, symbolic and structured representations.

| Visual State | Natural Language | Symbolic | Structured |
|---|---|---|---|
|  | "The player is at the upper-left, the box is to the right of the player, the target is below the player" | P_O, _X_, #__ | { 'player':[0,0], 'box':[1,1], 'target':[2,0], 'wall': [0,2] } |

**Table 3:** Performance comparison across different visual state representations. It shows that the representations are task-dependent.

| Visual State Representation | Sokoban | FrozenLake | PrimitiveSkill | | | | |
|---|---|---|---|---|---|---|---|
| | | | Place | Stack | Drawer | Align | Average |
| Natural-Lanaguage | 0.61 | 0.71 | 1.00 | 0.88 | 0.88 | 0.88 | 0.91 |
| Structured | 0.28 | 0.63 | 1.00 | 0.88 | 0.88 | 1.00 | 0.94 |
| Symbolic | 0.49 | 0.49 | – | – | – | – | – |

**Existing RL methods are inadequate for multi-turn VLM agents.** We also compare our VAGEN framework with different RL baselines. Vanilla PPO [11, 25] fails due to lack of observation token masking. For image/text-to-text models, learning from observation tokens is fundamentally incorrect as image tokens are not part of the model's generation process. For image/text-to-image/text models, learning from observation tokens might also be problematic because: (1) observation tokens are not generated by the agent's own policy, and (2) lengthy observation sequences can dominate the learning weight distribution. Group Relative Policy Optimization (GRPO) [43] with masking remains insufficient due to high trajectory diversity from scene change, requiring unaffordable sample sizes. Turn-level PPO with masking [13] underperforms because uniform advantage estimates for action tokens within a turn cannot capture individual token contributions to policy performance. These limitations motivate our VAGEN framework's design for effective VLM agent training.

## 3 How Can We Represent Internal Beliefs about the World?

To further understand visual state reasoning, we investigate how different visual state representations affect task performance. We consider three representations: Natural-Lanaguage, Symbolic, and Structured format, as shown in the example of Table 2. Specifically, during RL training, we prompt models to use the `WorldModeling` reasoning strategy and require them to output the specific format for the `<observation>` and `<prediction>` fields. We conducted these experiments in three tasks: FrozenLake, Sokoban, and PrimitiveSkill. In FrozenLake and Sokoban, we compare all three formats. The Natural-Lanaguage format consists of free-form textual descriptions. The Symbolic format uses environment-native grid-based symbols, and the Structured format requires the model to output a dictionary containing task-specific information such as players', targets' and boxes' positions, which are detailed in Appendix C. For PrimitiveSkill, we compare Natural-Lanaguage and Structured formats.

**Results and Insights.** Our experiments reveal a task-dependent trade-off in the choice of visual state representation. As shown in Table 3, the optimal format varies with the task nature. In FrozenLake and Sokoban, Natural-Lanaguage outperforms Symbolic and Structured formats. This is likely because the model lack sufficient prior knowledge to interpret symbolic layouts effectively, and structured outputs, when derived from image-only input, are noisy due to limited grounding capabilities. In these environments, the flexibility and familiarity of Natural-Lanaguage align better with the model's capabilities gained from pretraining stage. For PrimitiveSkill, Structured slightly outperforms Natural-Lanaguage. It's probably because we provided a structured object position list as prompts to the model, allowing the model to ground its understanding more precisely and facilitate more accurate next state prediction. Consequently, for our subsequent studies, we adopt Natural-Lanaguage as the default, general-purpose state representation, while specifically employing the Structured format for the PrimitiveSkill task.

# 4 Can World Modeling Help with Reward Shaping and Credit Assignment?

Recognizing the effectiveness of enforcing VLMs to explicitly reason as a world model on `StateEstimation` and `TransitionModeling`, we further explore to explicitly leverage these signals to inform reward structures and optimize the advantage estimation for the reinforcement learning framework.

## 4.1 WorldModeling Reward

Reward shaping is a common practice to guide the specific agent behavior. We aim to introduce a reward that supervises the agent's understanding of visual states. Specifically, we extract `<observation>` and `<prediction>` fields from the agent's response, compare them with the ground-truth visual states, and give a reward based on the matching score.

Our initial attempt use CLIP [44] based image-text similarity to calculate the reward. However, we found CLIP to be insufficiently sensitive to fine-grained spatial and geometric details, rendering the resulting reward signals unreliable.

To address this limitation, we adopt an LLM-as-a-Judge [21] approach. We try to get text-based ground-truth information about the visual state from the environments. For example, in Sokoban, we obtain 2D positions of the player, boxes, and targets; for FrozenLake, we extract 2D positions of the player, target, and holes; for PrimitiveSkill, we derive object names and their coordinates; and for Navigation, we calculate relative distances and directions from objects to the player.

With this text-based state information available, we compute a WorldModeling Reward by assessing the alignment between the agent's reasoning (in `<observation>` and `<prediction>`) and the ground truth states ($s_t$ and $s_{t+1}$). This is achieved through a hybrid evaluation protocol where an LLM-as-a-Judge either provides a direct judgment or first extracts structured information from the agent's text for a subsequent rule-based comparison (e.g., F1-score). The reward at each turn $t$ is defined as [5]:

$$r_t^{\text{reason}} = \beta_s \cdot \mathbb{I}_{\texttt{StateEstimation}}(\hat{s}_t, s_t) + \beta_w \cdot \mathbb{I}_{\texttt{TransitionModeling}}(\hat{s}_{t+1}, s_{t+1}),$$

where $\mathbb{I}$ is a generalized matching score (binary from direct judgment or a continuous, normalized score from rule-based metrics), and $\beta_s, \beta_w$ are reward coefficients. Details on the LLM-as-a-Judge prompts and evaluation protocols are available in Appendix D.1.

## 4.2 Bi-Level General Advantage Estimation (GAE)

The VAGEN-Base framework described in Section 2.2 reveals a key limitation when incorporating WorldModeling Reward: by aggregating all task rewards at the final token of a trajectory, it provides only trajectory-level feedback. This coarse signal is propagated backward via a single GAE calculation, making it difficult to assign credit for turn-specific successes or failures, which is especially critical for reinforcing step-by-step visual reasoning. To address this, we introduce **Bi-Level GAE**, a more granular credit assignment mechanism designed to deliver fine-grained, turn-level reward signals. As illustrated in Figure 3, this approach operates in two stages, introducing a turn-level discount factor $\gamma_{\text{turn}}$ for transitions between turns and a token-level discount factor $\gamma_{\text{token}}$ for tokens within a single action.

**Turn-level advantage estimation:** We compute an advantage estimate for each turn in the trajectory. For a given turn $t$, let $r_t$ be the total reward assigned to that turn (its composition will be detailed in Section 4.3). We define the turn-level TD-error $\delta_t^{\text{turn}}$ using the critic's value estimates $V_\phi$ at the end of each action sequence:

$$\delta_t^{\text{turn}} = r_t + \gamma_{\text{turn}} V_\phi(\bar{\tau}_{\leq a_{t+1}}) - V_\phi(\bar{\tau}_{\leq a_t}).$$

Here, $\bar{\tau}_{\leq a_t}$ denotes the full token prefix of the trajectory up to and including the action $a_t$. For the final turn $T - 1$, the next state value $V_\phi(\bar{\tau}_{\leq a_T})$ is considered zero. The turn-level advantage $A_t^{\text{turn}}$ is then calculated recursively using GAE:

$$A_t^{\text{turn}} = \delta_t^{\text{turn}} + \gamma_{\text{turn}} \lambda^{\text{turn}} A_{t+1}^{\text{turn}}.$$

---

[5]Please see Note 3, we regenerate current state belief and next state belief at each turn.

This backward pass is initialized with $A_{T-1}^{\text{turn}} = \delta_{T-1}^{\text{turn}}$, where $\lambda^{\text{turn}} \in [0, 1]$ is the GAE parameter for inter-turn credit assignment.

**Token-level advantage estimation:** After computing all turn-level advantages $\{A_0^{\text{turn}}, \ldots, A_{T-1}^{\text{turn}}\}$, we perform a second, inner GAE calculation for the tokens within each action $a_t$. The reward for any given token $\bar{\tau}_i$ within the action $a_t$ is defined as its KL-penalty, $r_i = r_i^{\text{KL}}$. The token-level TD-error and advantage are calculated for all tokens belonging to the action $a_t$:

$$\delta_{t,i}^{\text{token}} = r_{t,i}^{\text{KL}} + \gamma_{\text{token}} V_\phi(\bar{\tau}_{t,<i+1}) - V_\phi(\bar{\tau}_{t,<i}),$$

$$A_{t,i}^{\text{token}} = \delta_{t,i}^{\text{token}} + \gamma_{\text{token}} \lambda^{\text{token}} A_{t,i+1}^{\text{token}}.$$

The key step linking the two levels occurs here: the backward pass for the token-level advantages is initialized by setting the advantage of the **final token** of action $a_t$ to the pre-computed turn-level advantage, $A_t^{\text{turn}}$. This injects the turn-specific feedback at the end of the action and allows it to be propagated backward to all tokens that generated it. The detailed procedure is outlined in Algorithm 2 in the Appendix.

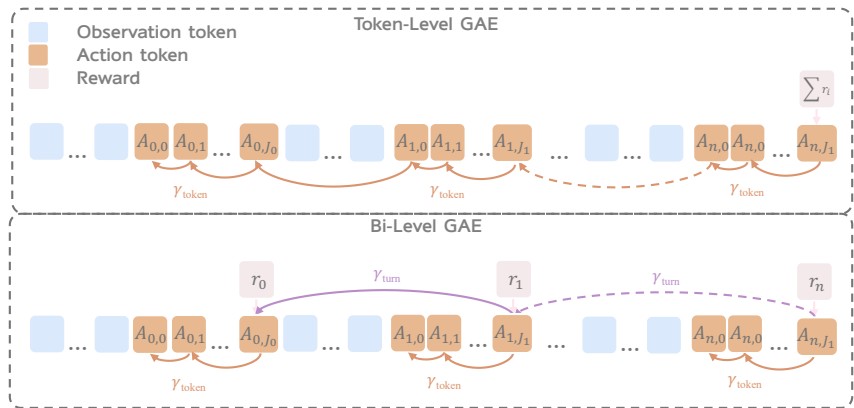

**Figure 3:** Token-Level GAE and Bi-Level GAE frameworks. Standard Token-Level GAE, where a single, sparse reward at the end of the trajectory is propagated backward token-by-token. Our proposed Bi-Level GAE assigns rewards at each turn. It first computes a turn-level advantage (purple arrows) before propagating that credit to the individual tokens within the turn (orange arrows), allowing for robust, hierarchical estimation for credit assignment.

### 4.3 VAGEN-Full Multi-Turn Reinforcement Learning Framework

By combining the structured reasoning strategies with our Bi-Level GAE mechanism, we introduce VAGEN-Full. In this setup, we train the agent using the `WorldModeling` reward to encourage explicit grounding and world modeling. The turn-level reward $r_t$ used in the Bi-Level GAE calculation is defined as a composite sum:

$$r_t = r_t^{\text{reason}} + r_t^{\text{format}} + R(s_t, a_t),$$

where $r_t^{\text{reason}}$ is a reward for the quality of the world model reasoning for visual states ($\hat{s}_t$ and $\hat{s}_{t+1}$), $r_t^{\text{format}}$ is the reward for adhering to the specified output structure (Section 2.4), and $R(s_t, a_t)$ is the sparse, task-specific reward from the environment. The rest of the training pipeline follows the VAGEN-Base procedure (Section 2.2), with the standard GAE module being replaced by the Bi-Level GAE for advantage estimation.

**Experiment Setup.** We now compare VAGEN-Base with VAGEN-Full across all tasks. The VAGEN-Base (Section 2.4) uses the `WorldModeling` reasoning strategy along with format and task-specific rewards. VAGEN-Full builds on this and incorporates World Modeling Reward and Bi-Level GAE, with the reward coefficients $\beta_s$ and $\beta_w$ set to 0.5. For SVG Reconstruction, only Bi-Level GAE is applied. Since SVG Reconstruction is a multi-turn reasoning task without world dynamics, we can only apply Bi-Level GAE configuration, and report VAGEN-BASE with Bi-Level GAE as the VAGEN-FULL results. Details regarding training time and token usage is given in Table 26.

**Results and Insights.** As shown in Table 1, VAGEN-Full achieves consistently better test-time performance across all tasks compared to VAGEN-Base (`WorldModeling`). This gap is especially prominent in PrimitiveSkill: although both methods reach similar training accuracy (Figure 4), VAGEN-Full

significantly outperforms VAGEN-Base on the test set. This suggests that `StateEstimation` and `TransitionModeling` improve the agent's ability to adapt to new scenes, leading to better robustness and generalization ability.

## 4.4 Ablations

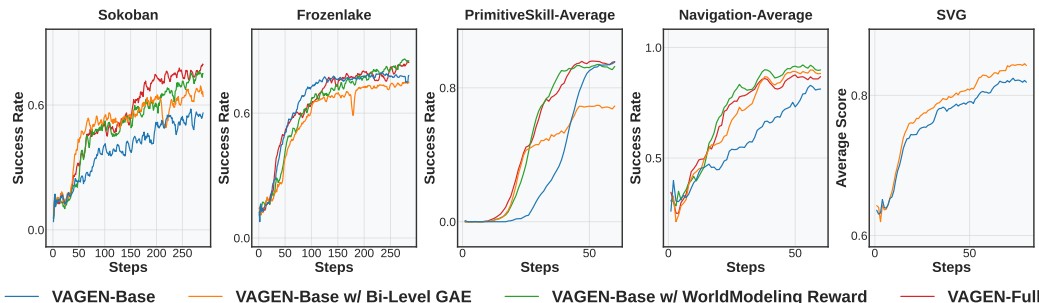

**Figure 4:** Training success rate for VAGEN-Base, VAGEN-Full and ablations. Since SVG reconstruction is a multi-turn reasoning task without world dynamics, we only apply the Bi-Level GAE configuration.

In this section, we study the independent contribution of Bi-Level GAE and WorldModeling Reward to VAGEN. We observe interesting patterns from Figure 4:

- **Bi-Level GAE alone** provides significant but inconsistent gains. Its performance is highly sensitive to reward sparsity and accuracy, which can lead to training instability in environments lacking dense and accurate intermediate rewards.
- **The WorldModeling Reward alone** consistently improves upon the baseline by providing a crucial learning signal for visual understanding. However, its effectiveness is limited by the coarse, trajectory-level credit assignment of standard RL.
- **VAGEN-Full** is the most stable among all methods and performs generally well on all tasks.

These observations verify that fine-grained credit assignment (from Bi-Level GAE) and high-quality reasoning supervision (from the WorldModeling Reward) are both essential to effectively improve VLM reasoning. Additional results comparing models (by size and family) and methods are reported in Table 25. We also provide case studies and a detailed analysis of agent behavior in Appendix E.

## 5 Related Work

Recent studies on visual reasoning have explored visual perception and grounding [45–49] and causal tracing in VLMs [50], and also examined visual information flow in single-turn QA scenarios [51–55]. However, maintaining visual state reasoning across multiple interaction turns, like building an internal world model, remains an underexplored challenge. Concurrently with our work, world modeling has been applied to code generation [56]. Our research addresses this gap by examining how VLMs maintain and update internal beliefs of visual states consistently during multi-turn interactions, focusing on improving reasoning of the environmental dynamics across consecutive turns with RL. We provide additional related work on RL and multi-turn agent training for LLMs and VLMs [5–7, 9, 12–14, 16, 20, 24, 40, 57–80] in Appendix F.

## 6 Conclusion and Limitations

We present a multi-turn RL framework that rewards reasoning to build an internal world model via explicit visual state reasoning over `StateEstimation` (grounding) and `TransitionModeling` (prediction). This encourages VLM agents to explore, model transition dynamics, and update beliefs across turns. We introduce a dense turn-level `WorldModeling` Reward that measures the accuracy of internal state simulations against ground truth, and Bi-Level GAE that first values an entire turn's reasoning before allocating credit to individual tokens, mitigating long-horizon credit assignment. VAGEN improves task performance and visual reasoning quality on agentic benchmarks. Limitations include reliance on a specific model family and a finite set of environments; future work will broaden architectures and incorporate supervised finetuning for multi-turn visual understanding.

## Acknowledgments and Disclosure of Funding

This work is in part supported by ONR N00014-23-1-2355, ONR YIP N00014-24-1-2117, ONR MURI N00014-24-1-2748, and ONR MURI N00014-22-1-2740.

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

# Appendix

## Table of Contents

**Table 4:** Summary of important notations used in the VAGEN framework.

| Notation | Symbol | Description |
|---|---|---|
| Observation | $o_t$ | The raw visual observation at turn $t$. |
| State | $s_t$ | Ground-truth environment state at turn $t$. |
| State (Belief) | $\hat{s}_t$ | VLM's belief of state $s_t$ at turn $t$. |
| Action | $a_t$ | The action tokens at $t$, $a_t = \langle z_t, a_t^e \rangle$, augmented with reasoning tokens $z_t$. |
| Action (Belief) | $\hat{a}_t$ | VLM's belief of action $a_t$ at turn $t$. |
| Action (Executable) | $a_t^e$ | The parsed, executable component of $a_t$. |
| Reward | $r_t$ | Scalar reward at turn $t$, $r_t = R(s_t, a_t)$ where $R$ is the reward function. |
| Trajectory | $\tau$ | Rollout $\tau = (s_0, o_0, a_0, r_0, \cdots, s_T, o_T)$. |
| Tokenized Trajectory | $\bar{\tau}$ | Token sequence for VLM training, concatenating encoded observations and actions. |
| Agent Policy | $\pi_\theta$ | Policy parameterized by a VLM with parameters $\theta$. |
| Critic / Value Func. | $V_\phi$ | State-value function parameterized by $\phi$. |
| Token Index | $i, j$ | $\bar{\tau}_i$: the $i$-th token. $\bar{\tau}_{i:j}$: tokens $i$–$j$. $\bar{\tau}_{<i}$: prefix up to $i - 1$. $\bar{\tau}_{t,i}$: the $i$-th token of the $t$-th turn. |
| Trajectory Reward | $R(\tau)$ | Sum of rewards over trajectory, $\sum_t R(s_t, a_t)$. |
| Advantage Estimate | $A$ | $A_i$: advantage for token $i$; $A_t^{\text{turn}}$: advantage for turn $t$; $A_{t,i}^{\text{token}}$: advantage for token $i$ in turn $t$. |
| Temporal-Difference Error | $\delta$ | $\delta_i$: TD-error for token $i$; $\delta_t^{\text{turn}}$: TD-error for turn $t$; $\delta_{t,i}^{\text{token}}$: TD-error for token $i$ in turn $t$. |
| Discount Factor | $\gamma$ | $\gamma \in [0, 1]$; $\gamma_{\text{token}}$: within-turn discount; $\gamma_{\text{turn}}$: across-turn discount. |

# Appendix

# A World Modeling for Visual State Reasoning via Multi-Turn RL

In this section, we detail our training pipeline, algorithm and system.

## A.1 Problem Formulation: VLM Agent Training under a POMDP Formulation

We choose to formulate our problem in a POMDP setting rather than a fully observable MDP for two main reasons.

**Inherent Partial Observability in Environments.** Many of the environments we study, such as the Navigation task, does not satisfy the assumptions of an MDP. In these environments, the agent does not have access to the complete, true state of the environment ($s_t$) at each timestep. Instead, it receives an observation $o_t$ (e.g., a first-person visual scene from its current vantage point) which constitutes only a partial view of $s_t$. To gain a more comprehensive understanding of the environment and to locate targets or critical information, the agent must actively explore, such as rotating its viewpoint or moving to a new location. This necessity to explore and gather information is a hallmark of POMDPs.

**Methodological Congruence with POMDPs.** We optimize policies over the full sequence of observations, actions, and reasoning steps by applying a teacher-forcing strategy during updates. This allows the model to leverage the entire trajectory as context for current decisions, ensuring consistency between rollout and update and making our method especially effective under partial observability.

## A.2 Multi-Turn Reinforcement Learning in VLM Agentic Tasks

Our multi-turn reinforcement learning framework for VLM agents is shown in Figure 5. At each turn $t$, the VLM agent receives the current observation $o_t$ (comprising a visual image and an optional textual prompt). Based on $o_t$ and its interaction history, the VLM generates a structured output $a_t$. This output $a_t$ includes an explicit reasoning component $z_t$ (e.g., `<think><observation>...<prediction>...</think>`) and an executable action component $a_t^e$ (e.g., `<answer>...</answer>`). The executable action $a_t^e$ is parsed and sent to the environment. The environment then transitions to a new state, providing the next observation $o_{t+1}$ and a scalar reward $r_t$. This cycle repeats for $N$ turns

to form a trajectory. The VLM's parameters are updated using reinforcement learning, specifically Proximal Policy Optimization (PPO), based on the collected trajectories. We leverage world modeling for visual state reasoning and design a training framework that integrates it with multi-turn trajectory optimization, as shown in Figure 6.

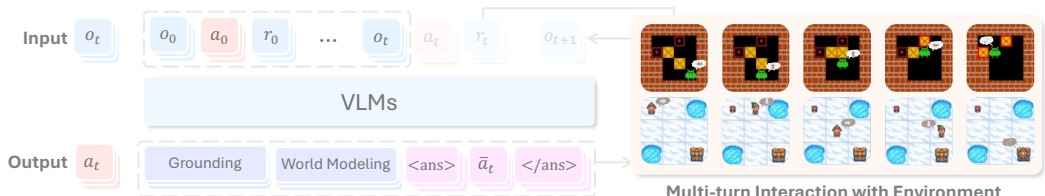

**Figure 5:** Multi-turn RL framework for VLM agents.

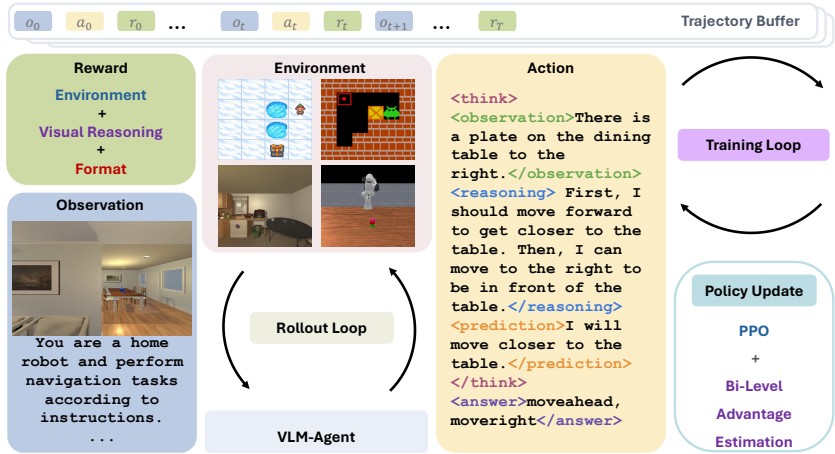

**Figure 6:** Overview of the VAGEN Framework. The framework operates in a dual-loop structure: an inner Rollout Loop for data collection and an outer Training Loop for policy optimization.

The detailed training algorithm for our multi-turn RL framework is presented in Algorithm 1.

The detailed reasoning templates is given in Table 5.

## A.3 VAGEN Framework

To support our experiments, we developed VAGEN, a framework built upon VeRL[81]. VAGEN specializes in multi-turn reinforcement learning (RL) training for Vision-Language Models (VLMs), integrating various RL algorithms and environments. We adopt *env-as-service* design, which decouples training from interacting with environments, enhances scalability.

## A.4 Environments and Tasks for VLM Agents

**Sokoban** The action space and hyperparameters are given in Table 6 and Table 7.

**Table 6:** Action space for the Sokoban environment.

| Name | Description |
|---|---|
| Up | Move the agent one cell upward on the grid. |
| Left | Move the agent one cell to the left on the grid. |
| Right | Move the agent one cell to the right on the grid. |
| Down | Move the agent one cell downward on the grid. |

---
**Algorithm 1** VAGEN-Base: Reinforcement Learning for VLM Agents
---

**Input:** Actor VLM $\pi_\theta$, Environment, Critic $V_\phi$, Reference VLM $\pi_{\text{ref}}$
**Hyperparameters:** Discount factor $\gamma$, GAE coefficient $\lambda$, PPO clip range $\varepsilon$, KL penalty coefficient $\beta$

---

**Phase 1: Trajectory Collection**
1: Initialize trajectory $\tau = []$; Sample initial state and observation $s_0, o_0$ from Environment
2: **for** $t = 0, \ldots, T-1$ **do**
3:     Sample action $a_t \sim \pi_\theta(\cdot|\bar{\tau}_{<a_t})$ autoregressively, where $\bar{\tau}_{<a_t}$ is the history.
4:     Parse executable action $a_t^{\text{e}}$ from $a_t$.
5:     Execute $a_t^{\text{e}}$ in Environment to get $s_{t+1}, o_{t+1}$ and task reward $R(s_t, a_t)$.
6:     Append $(s_t, o_t, a_t, R(s_t, a_t))$ to trajectory $\tau$.
7: **end for**
8: Construct training sequence $\bar{\tau} = [\mathcal{E}(o_0); \mathcal{E}(a_0); \cdots ; \mathcal{E}(o_{T-1}); \mathcal{E}(a_{T-1})]$.
9: Store old policy probabilities $\pi_{\text{old}}(\bar{\tau}_i|\bar{\tau}_{<i})$ for all generated tokens in $\bar{\tau}$.

---

**Phase 2: Advantage Estimation (Token-Level GAE)**
10: Let $M_i^{\text{loss}}$ be a mask that is 1 for action tokens and 0 for observation tokens.
11: Calculate total trajectory task reward $R(\tau) = \sum_{t=0}^{T-1} R(s_t, a_t)$.
12: **for** each action token index $i = 0, \ldots, |\bar{\tau}| - 1$ **do**
13:     Calculate KL penalty: $r_i^{\text{KL}} = -\beta \cdot \text{KL}\left[\pi_\theta(\cdot|\bar{\tau}_{<i}) \,\|\, \pi_{\text{ref}}(\cdot|\bar{\tau}_{<i})\right]$.
14:     Initialize token reward $r_i = r_i^{\text{KL}}$.
15: **end for**
16: Add trajectory reward to the final action token: $r_{|\bar{\tau}|-1} \leftarrow r_{|\bar{\tau}|-1} + R(\tau)$.
17: Initialize advantage $A_{|\bar{\tau}|} = 0$.
18: **for** $i = |\bar{\tau}| - 1, \ldots, 0$ (backwards) **do**
19:     Find the least index $j$ of an action token where $j > i$
20:     Compute TD-error: $\delta_i = r_i + \gamma V_\phi(\bar{\tau}_{<j}) - V_\phi(\bar{\tau}_{<i})$. (Assume $V_\phi(\text{terminal}) = 0$)
21:     Compute advantage: $A_i = \delta_i + \gamma\lambda A_j$.
22:     Compute return for critic update: $G_i = A_i + V_\phi(\bar{\tau}_{<i})$.
23: **end for**

---

**Phase 3: Policy Update with PPO**
24: **for** each action token index $i = 0, \ldots, |\bar{\tau}| - 1$ **do**
25:     Compute probability ratio: $u_i(\theta) = \frac{\pi_\theta(\bar{\tau}_i|\bar{\tau}_{<i})}{\pi_{\text{old}}(\bar{\tau}_i|\bar{\tau}_{<i})}$.
26: **end for**
27: Compute PPO objective:
     $J^{\text{PPO}}(\theta) = \frac{1}{\sum M_i^{\text{loss}}} \sum_i M_i^{\text{loss}} \cdot \min\left(u_i(\theta)A_i, \text{clip}(u_i(\theta), 1 - \varepsilon, 1 + \varepsilon)A_i\right)$.
28: Compute critic loss:
     $J^{\text{Critic}}(\phi) = \frac{1}{\sum M_i^{\text{loss}}} \sum_i M_i^{\text{loss}} \cdot (V_\phi(\bar{\tau}_{<i}) - G_i)^2$.
29: Update parameters $\theta$ and $\phi$ using their respective gradients.

---

**Table 7:** Hyperparameters for the Sokoban environment.

| Name | Value | Description |
|------|-------|-------------|
| dim_room | $(6, 6)$ | Dimensions of the Sokoban grid |
| max_steps | 100 | Maximum number of steps allowed per episode |
| num_boxes | 1 | Number of boxes to be pushed onto targets |
| min_actions_to_succeed | 5 | Minimum number of actions required to solve the puzzle |
| max_actions_per_step | 3 | Maximum number of actions the agent can take per turn |
| max_turns | 3 | Maximum number of turns the agent can interact with the environment |

**FrozenLake** The action space and hyperparameters are given in Table 8 and Table 9.

**Table 5:** Reasoning strategy formats.

| Strategy | Format |
|---|---|
| NoThink | `<answer>`$a_t^e$`</answer>` |
| FreeThink | `<think>`$z_t$`</think>` |
| | `<answer>`$a_t^e$`</answer>` |
| StateEstimation | `<think>` |
| | `<observation>`$\hat{s}_t$`</observation>` |
| | `<reasoning>`$\hat{a}_t$`</reasoning>` |
| | `</think>` |
| | `<answer>`$a_t^e$`</answer>` |
| TransitionModeling | `<think>` |
| | `<reasoning>`$\hat{a}_t$`</reasoning>` |
| | `<prediction>`$\hat{s}_{t+1}$`</prediction>` |
| | `</think>` |
| | `<answer>`$a_t^e$`</answer>` |
| WorldModeling | `<think>` |
| | `<observation>`$\hat{s}_t$`</observation>` |
| | `<reasoning>`$\hat{a}_t$`</reasoning>` |
| | `<prediction>`$\hat{s}_{t+1}$`</prediction>` |
| | `</think>` |
| | `<answer>`$a_t^e$`</answer>` |

**Table 8:** Action space for the FrozenLake environment.

| Name | Description |
|---|---|
| Up | Move the agent one cell upward on the grid. |
| Left | Move the agent one cell to the left on the grid. |
| Right | Move the agent one cell to the right on the grid. |
| Down | Move the agent one cell downward on the grid. |

**Table 9:** Hyperparameters for the FrozenLake environment.

| Name | Value | Description |
|---|---|---|
| desc | None | Environment map layout (if None, randomly generated) |
| is_slippery | False | Whether the frozen tiles are slippery |
| size | 4 | Size of the square grid |
| max_actions_per_step | 3 | Maximum number of actions the agent can take per turn |
| min_actions_to_succeed | 5 | Minimum number of actions required to reach the goal |
| max_turns | 3 | Maximum number of turns the agent can interact with the environment |

**Navigation** The action space and hyperparameters are given in Table 10 and Table 11.

**Table 10:** Action space for the Navigation environment.

| Name | Description |
|------|-------------|
| MoveAhead | Move forward by some distance |
| MoveBack | Move backward by some distance |
| MoveRight | Move rightward by some distance |
| MoveLeft | Move leftward by some distance |
| RotateRight | Rotate to the right by 90 degrees |
| RotateLeft | Rotate to the left by 90 degrees |
| LookUp | Tilt the camera upward by 30 degrees |
| LookDown | Tilt the camera downward by 30 degrees |

**Table 11:** Hyperparameters for the Navigation environment.

| Name | Value | Description |
|------|-------|-------------|
| resolution | 255 | Resolution of the rendered images |
| down_sample_ratio | 1.0 | Ratio for down-sampling images |
| fov | 100 | Field of view angle in degrees |
| multiview | False | Whether to use multiple camera views |
| max_actions_per_step | 5 | Maximum number of actions the agent can take per turn |
| success_threshold | 1.5 | Threshold for considering task successful |
| step_length | 0.5 | Distance traveled in a single movement action |
| max_turns | 4 | Maximum number of turns the agent can interact with the environment |

**PrimitiveSkill** The action space and hyperparameters are given in Table 12 and Table 13.

**Table 12:** Action space for the PrimitiveSkill environment.

| Name | Description |
|------|-------------|
| pick(x, y, z) | Grasp an object located at position (x, y, z) in the robot's workspace |
| place(x, y, z) | Place the object currently held by the robot's gripper at the target position (x, y, z) |
| push(x1, y1, z1, x2, y2, z2) | Push an object from position (x1, y1, z1) to position (x2, y2, z2) |

**Table 13:** Hyperparameters for the PrimitiveSkill environment.

| Name | Value | Description |
|------|-------|-------------|
| max_actions_per_step | 2 | Maximum number of actions the agent can take per turn |
| max_turns | 3 | Maximum number of turns the agent can interact with the environment |

**SVG Reconstruction** The action space and hyperparameters are given in Table 14 and Table 15.

**Table 14:** Action space for the SVG Reconstruction environment.

| Name | Description |
|------|-------------|
| SVG Code | Open text format allowing for the specification of SVG markup code |

**Table 15:** Hyperparameters for the SVG Reconstruction environment.

| Name | Value | Description |
|------|-------|-------------|
| dataset_name | starvector/svg-icons-simple | Dataset used for SVG examples |
| max_turns | 2 | Maximum number of turns the agent can interact with the environment |

## A.5 Reward Assignment

**Sokoban** The reward structure for the Sokoban environment is presented in Table 16.

**Table 16:** Reward structure for the Sokoban environment.

| Reward Type | Value | Description |
|-------------|-------|-------------|
| Success reward | 10 | Awarded when all boxes are placed on target locations |
| Failure penalty | -0.1 | Applied each step when the task is not completed |
| Box placement reward | 1 | Granted for each box pushed onto a target location |
| Format reward | 0.5 | Provided at each turn to encourage visual state reasoning |
| Grounding reward weight | 0.5 | Weight applied to `StateEstimation` reward |
| World modeling reward weight | 0.5 | Weight applied to `TransitionModeling` reward |

**FrozenLake** The reward structure for the FrozenLake environment is presented in Table 17.

**Table 17:** Reward structure for the FrozenLake environment.

| Reward Type | Value | Description |
|-------------|-------|-------------|
| Success reward | 10 | Awarded when the agent reaches the goal position |
| Failure penalty | -0.1 | Applied each step when the task is not completed |
| Format reward | 0.5 | Provided at each turn to encourage visual state reasoning |
| Grounding reward weight | 0.5 | Weight applied to `StateEstimation` reward |
| World modeling reward weight | 0.5 | Weight applied to `TransitionModeling` reward |

**Navigation** The reward structure for the Navigation environment is presented in Table 18.

**Table 18:** Reward structure for the Navigation environment.

| Reward Type | Value | Description |
|-------------|-------|-------------|
| Success reward | 10 | Awarded when the agent reaches the goal location |
| Failure penalty | -0.1 | Applied each step when the task is not completed |
| Format reward | 0.5 | Provided at each turn to encourage visual state reasoning |
| Grounding reward weight | 0.5 | Weight applied to `StateEstimation` reward |
| World modeling reward weight | 0.5 | Weight applied to `TransitionModeling` reward |

**PrimitiveSkill** The reward structure for the PrimitiveSkill environment is presented in Table 19.

**Table 19:** Reward structure for the PrimitiveSkill environment.

| Reward Type | Value | Description |
|---|---|---|
| Success reward | 10 | Awarded when the manipulation task is completed |
| Failure penalty | -0.1 | Applied each step when the task is not completed |
| Stage-based reward | $(\text{stage} + 1) \times 2$ | Granted upon completing key manipulation subgoals (where stage is the highest successfully completed stage) |
| Format reward | 0.5 | Provided at each turn to encourage visual state reasoning |
| Grounding reward weight | 0.5 | Weight applied to `StateEstimation` reward |
| World modeling reward weight | 0.5 | Weight applied to `TransitionModeling` reward |

**SVG** The reward structure for the SVG Reconstruction environment is presented in Table 20.

**Table 20:** Reward structure for the SVG Reconstruction environment.

| Reward Type | Value | Description |
|---|---|---|
| Image similarity | Variable | Weighted DreamSim [82] and DINO [83] scores measuring similarity between generated and target images |
| Format reward | 0.5 | Provided at each turn to encourage visual state reasoning |
| Grounding reward weight | 0.5 | Weight applied to `StateEstimation` reward |
| World modeling reward weight | 0.5 | Weight applied to `TransitionModeling` reward |
| DreamSim weight | 5.0 | Scaling factor applied to DreamSim similarity scores |
| Dino weight | 0.0001 | We only use DreamSim score for reward |

### A.6 Evaluation Metrics

We employ a range of metrics to evaluate agent performance across our diverse task environments. For trajectory-based evaluation, we define several key functions over a trajectory $\tau$:

- $f(\tau) \in \{0, 1\}$: A binary success indicator function that equals 1 if the trajectory $\tau$ successfully completes the task, and 0 otherwise
- $g(\tau) \in [0, 1]$: The DreamSim similarity score between the target image and the final generated image in trajectory $\tau$
- $h(\tau) \in [0, 1]$: The DINO similarity score between the target image and the final generated image in trajectory $\tau$

**Metrics for Task-Completion Environments** For Sokoban, FrozenLake, Navigation, and PrimitiveSkill environments, we use the average success rate over a dataset $\mathcal{D}$ of test trajectories:

$$\text{Success Rate} = \mathbb{E}_{\tau \sim \mathcal{D}}[f(\tau)] \tag{1}$$

A trajectory is considered successful ($f(\tau) = 1$) when the agent completes the specific task objectives for each environment.

**Metrics for SVG Reconstruction** For the SVG Reconstruction task, where success is measured by the similarity between generated and target images rather than discrete goal achievement, we report:

$$\text{DreamSim Score} = \mathbb{E}_{\tau \sim \mathcal{D}}[g(\tau)] \tag{2}$$
$$\text{DINO Score} = \mathbb{E}_{\tau \sim \mathcal{D}}[h(\tau)] \tag{3}$$

These complementary metrics capture different aspects of visual similarity:

- **DreamSim Score**: Measures perceptual similarity between images using a pre-trained diffusion model [82].
- **DINO Score**: Based on self-supervised visual features from a small version of the DINO model [83].

# B What Can We Reason About Visual States?

## B.1 Bottleneck of Off-the-Shelf VLMs on Agentic Tasks

To understand the capabilities and limitations of existing VLMs on agentic tasks, we evaluate several state-of-the-art off-the-shelf VLMs across our proposed tasks. The prompts used for evaluating these models follow the `FreeThink` reasoning strategy format: `<think>`$t_t$`</think> <answer>` $a_t^e$ `</answer>`.

---

**Sokoban Prompt for evaluating off-the-shelf VLMs**

```
System
You are a Sokoban solver.
Sokoban Quick Guide
Goal: Push all boxes onto targets.
Symbols (If image is provided there are no symbols):
# Wall | _ Floor | O Target | X Box | P You | * Box on Target | S You on
Target
Rules:
1. Push boxes (can't pull).
2. Avoid walls.
Actions you can take: Left, Down, Right, Up.
You can take up to 3 action(s) at a time, separated by ,.
You should first give your reasoning, and then your answer.
Your response should be in the format of:
<think>...</think><answer>...</answer>
e.g. <think><reasoning>The box is one step below me, and the target is two
steps below me, I need to go down then push the box down to the
target.</reasoning></think><answer>Down,Down</answer>
User
[Initial Observation]:
<image>
Decide your next action(s).
You can take up to 3 action(s) at a time, separated by ,.
You should first give your reasoning, and then your answer.
Your response should be in the format of:
<think>...</think><answer>...</answer>
Assistant
<think>Since there are no boxes or targets visible, the first step is to
move the character (P) to the wall to see if it will push the wall and open
a path. This might also help in accessing more space to maneuver.</think>
<answer>Left, Up, Right</answer>
rest of the turns ...
```

---

**FrozenLake Prompt for evaluating off-the-shelf VLMs**

```
System
You are a FrozenLake solver.
FrozenLake Quick Guide
Goal: Reach the goal (G).
Symbols (If image is provided there are no symbols):
_ Frozen | O Hole | G Goal | P Player | X Player fell into hole | * Player
on goal
Rules:
1. Avoid falling into holes.
2. Frozen tiles are slippery, you may move perpendicular to your intended
direction.
Actions you can take: Left, Down, Right, Up.

You can take up to 3 action(s) at a time, separated by ,.
You should first give your reasoning, and then your answer.
```

```
Your response should be in the format of:
<think>...</think><answer>...</answer>
e.g. <think>I can see the target is on my down left, I should go down then
left to reach the target</think><answer>Down,Left</answer>
```
User
```
[Initial Observation]:
<image>
Decide your next action(s).

You can take up to 3 action(s) at a time, separated by ,.
You should first give your reasoning, and then your answer.
Your response should be in the format of:
<think>...</think><answer>...</answer>
```
Assistant
```
<think>There is a present at the bottom right, which is likely the goal.
Since the player is near the present and the goal is usually at the bottom
right, I should move downwards to reach it.</think><answer>Down, Down,
Down</answer>
```
rest of the turns ...

---

Navigation Prompt for evaluating off-the-shelf VLMs

System
```
You are a home robot and perform navigation tasks according to instructions.
Actions you can take: moveahead, moveback, moveright, moveleft, rotateright,
rotateleft, lookup, lookdown.
moveahead: Move forward by some distance
moveback: Move backward by some distance
moveright: Move rightward by some distance
moveleft: Move leftward by some distance
rotateright: Rotate to the right by 90 degrees
rotateleft: Rotate to the left by 90 degrees
lookup: Tilt the camera upward by 30 degrees
lookdown: Tilt the camera downward by 30 degrees
Rewards:
Format correct: +0.5
Achieve the human instruction: +10.0
The instruction will be provided with each observation. Look at the image
carefully and navigate to complete the instruction.
Hints:
1. You can take multiple actions at a time, in most cases, if you find the
target object is far away from you, you can call moveahead, moveleft and
move right multiple times.
2. If you find yourself seems to be stuck, you can lookdown to see if
there's any object above or below you, you can also rotate to see if there's
any object behind you.
Example:
Round 1:
image_1
<think>I can see the garbage can in the upper left corner of the image, next
to the kitchen sink. To move there, we can go forward-left, but since
there's a kitchen counter directly ahead, we should go left first. Following
the strategy, I can go by first moving leftward.</think>
<answer>moveleft, moveleft</answer>
Round 2:
Env_feedback: Last action is executed successfully.
image_2
<think>From the secene, I see that by moving leftward, we are getting closer
to the garbage can. Now, the garbage can is in front of me, slightly to the
left. And there's a large area ahead of us. Following the strategy, I can go
by first moving forward then moving leftward.</think>
```

```
<answer>moveahead, moveahead,moveahead,moveleft</answer>
Round 3:
Env_feedback: Last action is executed successfully.
image_3
<think>From the image we can see the garbage can is very close to us, still
to our front-left. Moving leftward might be blocked but i can see that there
is still space in front of me to get closer to the garbage can. Following
the strategy, we can take about two steps forward then one step left to
reach the garbage can.</think>
<answer>moveahead, moveahead,moveleft</answer>
Round 4:
Env_feedback: Success
You can take up to 5 action(s) at a time, separated by ','.
You should first give your thought process, and then your answer.
Your response should be in the format of:
<think>...</think><answer>...</answer>
e.g. <think>I can see from the sight the target object is right in the top
left of me, I will move forward, then move left to access
it.</think><answer>moveahead,moveahead,moveahead,moveleft,moveleft</answer>
User
[Initial Observation]:
<image>
Human Instruction: I need to dispose of some trash properly. Please navigate
to that object and stay near it.
Decide your next action(s).
You can take up to 5 action(s) at a time, separated by ','.
You should first give your thought process, and then your answer.
Your response should be in the format of:
<think>...</think><answer>...</answer>
Assistant
<think>From the image, I can see the trash can is in the bottom-right
corner. I need to navigate towards it. I will first move forward and then
turn to the right to reach it.</think><answer>moveahead, rotateright</answer>
rest of the turns ...
```

PrimitiveSkill Prompt for evaluating off-the-shelf VLMs

```
System
You are an AI assistant controlling a Franka Emika robot arm. Your goal is
to understand human instructions and translate them into a sequence of
executable actions for the robot, based on visual input and the instruction.

Action Space Guide
You can command the robot using the following actions:

1. pick(x, y, z) # To grasp an object located at position(x,y,z) in the
robot's workspace.
2. place(x, y, z) # To place the object currently held by the robot's
gripper at the target position (x,y,z).
3. push(x1, y1, z1, x2, y2, z2) # To push an object from position (x1,y1,z1)
to (x2,y2,z2).

Hints:
1. The coordinates (x, y, z) are in millimeters and are all integers.
2. Please ensure that the coordinates are within the workspace limits.
3. The position is the center of the object, when you place, please consider
the volume of the object. It's always fine to set z much higher when placing
an item.
4. We will provide the object positions to you, but you need to match them
to the object in the image by yourself. You're facing toward the negative
```

x-axis, and the negative y-axis is to your left, the positive y-axis is to
your right, and the positive z-axis is up.

Examples:
round1:
image1
Human Instruction: Put red cube on green cube and yellow cube on left target
Object positions:
[(62,-55,20),(75,33,20),(-44,100,20),(100,-43,0),(100,43,0)]
Reasoning: I can see from the picture that the red cube is on my left and
green cube is on my right and near me.
Since I'm looking toward the negative x axis, and negative y-axis is to my
left, (62,-55,20) would be the position of the red cube, (75,33,20) would be
the position of the green cube and (-44,100,20) is the position of the
yellow cube.
Also the (100,-43,0) would be the position of the left target, and
(100,43,0) would be the porition of the right target.
I need to pick up red cube first and place it on the green cube, when
placing, I should set z much higher.
Anwer: pick(62,-55,20)|place(75,33,50)
round2:
image2
Human Instruction: Put red cube on green cube and yellow cube on left target
Object positions:
[(75,33,50),(75,33,20),(-44,100,20),(100,-43,0),(100,43,0)]
Reasoning: Now the red cube is on the green cube, so I need to pick up the
yellow cube and place it on the left target.
Anwer: pick(-44,100,20)|place(100,-43,50)

You can take up to 2 action(s) at a time, separated by |.
You should first give your thought process, and then your answer.
Your response should be in the format of:
<think>...</think><answer>...</answer>
e.g. e.g. <think>I need to pick the red_cube_pos at (10,20,30) and place it
on the green_block_pos at
(50,60,40).</think><answer>pick(10,20,30)|place(50,60,70)</answer>
User
[Initial Observation]:
<image>
Human Instruction: Please align the cubes in the y-axis, which means the
x-coordinates of both cubes should be 0 (+-10mm)
x_workspace_limit: (-100, 150)
y_workspace_limit: (-200, 200)
z_workspace_limit: (10, 200)
Object positions:
[(129, -119, 20), (108, 124, 20)]
Other information:
No other information needed
Decide your next action(s).
You can take up to 2 action(s) at a time, separated by |.
You should first give your thought process, and then your answer.
Your response should be in the format of:
<think>...</think><answer>...</answer>
Assistant
<think>First, I need to pick up the red cube at position (129, -119, 20) and
place it at position (0, -119, 20).</think><answer>pick(129, -119,
20)|place(0, -119, 50)</answer>
rest of the turns ...

```
System
You are a precise SVG code generator.

SVG Quick Guide
Goal: Transform the provided image into precise SVG code that replicates the
image.

Process:
1. First analyze the image carefully, identifying distinct visual elements
2. Identify colors, dimensions, positions, and relationships between elements
3. Generate accurate SVG code that reproduces the image, you cam use path
for better shape

Rewards:
- Overall visual similarity: +5.0
- Structural accuracy: +10.0
Example:
<think>I can see the image contains a red circle and a blue rectangle. The
circle is positioned at the top-left, while the rectangle is at the
bottom-right.</think>
<answer><svg viewBox="0 0 100 100" xmlns="http://www.w3.org/2000/svg">
  <circle cx="25" cy="25" r="15" fill="red" />
  <rect x="60" y="60" width="30" height="20" fill="blue" />
</svg></answer>
You can take up to 1 action(s) at a time, separated by ~~.
You should first give your thought process, and then your answer.
Your response should be in the format of:
<think>...</think><answer>...</answer>
e.g. <think>I can see the image contains a red circle and a blue rectangle.
The circle is positioned at the top-left, while the rectangle is at the
bottom-right.</think>
<answer><svg viewBox="0 0 100 100" xmlns="http://www.w3.org/2000/svg">
  <circle cx="25" cy="25" r="15" fill="red" />
  <rect x="60" y="60" width="30" height="20" fill="blue" />
</svg></answer>
User
[Initial Observation]:
<image>
Please carefully observe the image, and generate SVG code that reproduces it
as accurately as possible.
Decide on your SVG code.

You can take up to 1 action(s) at a time, separated by ~~.
You should first give your thought process, and then your answer.
Your response should be in the format of:
<think>...</think><answer>...</answer>
Assistant
<think>Based on the image, I can see a circular shape with the letters "kr"
inside it. The letters are positioned in the center of the circle.</think>
<answer><svg viewBox="0 0 100 100" xmlns="http://www.w3.org/2000/svg">
  <circle cx="50" cy="50" r="40" fill="none" stroke="black" stroke-width="2"
/>
  <text x="50" y="70" font-size="40" font-family="Arial"
fill="black">kr</text>
</svg></answer>
rest of the turns ...
```

The information of evaluated VLMs with corresponding hyperparameters are provided in Table 21.

**Table 21:** Off-the-shelf VLMs for evaluation.

| Model Name | Model ID | Provider | Max Tokens | Temperature |
|---|---|---|---|---|
| VLM-R1-3B | omlab/VLM-R1-Qwen2.5VL-3B-Math-0305 | omlab | 150/400 | 0.7 |
| Qwen2.5-VL-3B | Qwen/Qwen2.5-VL-3B-Instruct | Qwen | 150/400 | 0.7 |
| Qwen2.5-VL-7B | Qwen/Qwen2.5-VL-7B-Instruct | Qwen | 150/400 | 0.7 |
| Qwen2.5-VL-72B | Qwen/Qwen2.5-VL-72B-Instruct | Qwen | 150/400 | 0.7 |
| GPT-4o | gpt-4o | openai | 150/400 | 0.7 |
| Gemini 2.0 | gemini-2.0-flash | gemini | 150/400 | 0.7 |
| Claude 3.7 Sonnet | claude-3-7-sonnet-20250219 | claude | 150/400 | 0.7 |

## B.2  Reasoning in Multi-turn RL Training

Following the training methodology described in Section 2.2, we use reinforcement learning to train Qwen2.5-VL-3B across all reasoning strategies presented in Table 5. Our experiments are conducted on servers equipped with 8×H100 GPUs, 104 CPUs, and 1.7TB of memory. Each server can run two experiments at the same time, with each training session requiring approximately 4-8 hours to complete.

The training hyperparameters used in our experiments are detailed in Table 22.

**Table 22:** Multi-turn RL training hyperparameters.

| Parameter | Value | Description |
|---|---|---|
| **Rollout Phase** | | |
| Top-p | 0.95 | Nucleus sampling parameter for action generation |
| Temperature | 0.7 | Sampling temperature for controlling randomness |
| **Update Phase** | | |
| Advantage Estimator | masked_gae | Generalized Advantage Estimation with masking |
| Actor Model | Qwen/Qwen2.5-VL-3B-Instruct | Pre-trained model used for actor initialization |
| Critic Model | Qwen/Qwen2.5-VL-3B-Instruct | Pre-trained model used for critic initialization |
| $\gamma_{token}$ | 1.0 | Discount factor for token-wise advantage calculation |
| KL Penalty Coefficient ($\beta$) | 0.001 | Coefficient for KL divergence penalty in PPO objective |
| Actor Learning Rate | 1e-6 | Learning rate for the actor network |
| Critic Learning Rate | 1e-5 | Learning rate for the critic network |
| Train Batch Size | 128 | Total batch size for training |
| PPO Mini Batch Size | 32 | Mini-batch size for PPO updates |

We also release the prompts used in the multi-turn RL training process. For different reasoning strategies, the structures of the prompts are largely identical, with minor differences in response formats. To avoid redundancy, we provide only the `WorldModeling` prompts here. For `FreeThink` prompts, please refer to Appendix B.1. For other reasoning strategy prompts, please see our provided code or infer them from the existing prompts.

---

**Sokoban Training Prompt for `WorldModeling` Strategy**

`System`
```
You are a Sokoban solver.
Sokoban Quick Guide
Goal: Push all boxes onto targets.
Symbols (If image is provided there are no symbols):
# Wall | _ Floor | O Target | X Box | P You | * Box on Target | S You on
Target
Rules:
1. Push boxes (can't pull).
2. Avoid walls.
Actions you can take: Left, Down, Right, Up.
You can take up to 3 action(s) at a time, separated by ,.
```

You should first give the description of your observation, then your
reasoning, then predict the next state, and finally your answer.
Your response should be in the format of:
<think><observation>...</observation><reasoning>...</reasoning>
<prediction>...</prediction></think><answer>...</answer>
e.g. <think><observation>The box is below the player and the target is below
the box</observation><reasoning>I need to go down then push the box down to
the target</reasoning><prediction>The player will be above the box, the
target and box will be at the same
place.</prediction></think><answer>Down,Down</answer>
User
[Initial Observation]:
<image>
Decide your next action(s).
You can take up to 3 action(s) at a time, separated by ,.
You should first give the description of your observation, then your
reasoning, then predict the next state, and finally your answer.
Your response should be in the format of:
<think><observation>...</observation><reasoning>...</reasoning>
<prediction>...</prediction></think><answer>...</answer>
Assistant
<think><observation>The player is at the bottom of the screen, and there is
a box to the right of the player. The target is to the left of the
box.</observation><reasoning>The player needs to push the box to the target
to complete the goal.</reasoning><prediction>The player will push the box to
the target, moving up, down, and to the
left.</prediction></think><answer>Up, Down, Left</answer>
rest of the turns ...

---

**FrozenLake Training Prompt for** `WorldModeling` **Strategy**

System
You are a FrozenLake solver.
FrozenLake Quick Guide
Goal: Reach the goal (G).
Symbols (If image is provided there are no symbols):
_ Frozen | O Hole | G Goal | P Player | X Player fell into hole | * Player
on goal
Rules:
1. Avoid falling into holes.
2. Frozen tiles are slippery, you may move perpendicular to your intended
direction.
Actions you can take: Left, Down, Right, Up.

You can take up to 3 action(s) at a time, separated by ,.
You should first describe the observation, then your reasoning, then predict
the next state, and finally your answer.
Your response should be in the format of:
<think><observation>...</observation><reasoning>...</reasoning>
<prediction>...</prediction></think><answer>...</answer>
e.g. <think><observation>The player is on the above the
target</observation><reasoning>I should go down then left to reach the
target</reasoning><prediction>The player will reach the
target</prediction></think><answer>Down,Left</answer>
User
[Initial Observation]:
<image>
Decide your next action(s).

You can take up to 3 action(s) at a time, separated by ,.

You should first describe the observation, then your reasoning, then predict the next state, and finally your answer.
Your response should be in the format of:
<think><observation>...</observation><reasoning>...</reasoning>
<prediction>...</prediction></think><answer>...</answer>
Assistant
<think><observation>The player is on the right side of the grid.</observation><reasoning>The player is on the right side of the grid, which is indicated by the position on the grid.</reasoning><prediction>The player will move to the left or down.</prediction></think><answer>Left, Left, Down</answer>
rest of the turns ...

---

Navigation Training Prompt for `WorldModeling` Strategy

System
You are a home robot and perform navigation tasks according to instructions.
Actions you can take: moveahead, moveback, moveright, moveleft, rotateright, rotateleft, lookup, lookdown.
moveahead: Move forward by some distance
moveback: Move backward by some distance
moveright: Move rightward by some distance
moveleft: Move leftward by some distance
rotateright: Rotate to the right by 90 degrees
rotateleft: Rotate to the left by 90 degrees
lookup: Tilt the camera upward by 30 degrees
lookdown: Tilt the camera downward by 30 degrees
Rewards:
Format correct: +0.5
Achieve the human instruction: +10.0
The instruction will be provided with each observation. Look at the image carefully and navigate to complete the instruction.
Hints:
1. You can take multiple actions at a time, in most cases, if you find the target object is far away from you, you can call moveahead, moveleft and move right multiple times.
2. If you find yourself seems to be stuck, you can lookdown to see if there's any object above or below you, you can also rotate to see if there's any object behind you.
Example:
Round 1:
image_1
<think><observation>There is a garbage can in the upper left corner of the image, next to the kitchen sink. To move there, we can go forward-left, but since there's a kitchen counter directly ahead, we should go left first.</observation><reasoning>Following the strategy, I can go by first moving leftward.</reasoning><prediction>I will be infront of the garbage</prediction></think>
<answer>moveleft, moveleft</answer>
Round 2:
Env_feedback: Last action is executed successfully.
image_2
<think><observation>From the secene, I see that by moving leftward, we are getting closer to the garbage can. Now, the garbage can is in front of me, slightly to the left. And there's a large area ahead of us.</observation><reasoning>Following the strategy, I can go by first moving forward then moving leftward.</reasoning><prediction>I will be closer to the garbage</prediction></think>
<answer>moveahead, moveahead,moveahead,moveleft</answer>
Round 3:
Env_feedback: Last action is executed successfully.

```
image_3
<think><observation>From the image we can see the garbage can is very close
to us, still to our front-left. Moving leftward might be blocked but i can
see that there is still space in front of me to get closer to the garbage
can.</observation><reasoning>Following the strategy, we can take about two
steps forward then one step left to reach the garbage
can.</reasoning><prediction>I will reach the garbage</prediction></think>
<answer>moveahead, moveahead,moveleft</answer>
Round 4:
Env_feedback: Success
You can take up to 5 action(s) at a time, separated by ','.
You should first give your thought process with the your observation,
reasoning, and prediction of next state, then your answer.
Both the observation and prediction should describe what you see or expect
to see in the environment.
Your response should be in the format of:
<think><observation>...</observation><reasoning>...</reasoning>
<prediction>...</prediction></think><answer>...</answer>
e.g. <think><observation>I am at the entrance of a bedroom. There is a bed
to the left, a desk with a lamp on the right, and a closet straight ahead.
The target object, a book, appears to be on the
desk.</observation><reasoning>I need to move toward the desk to reach the
book. I'll turn right and move forward.</reasoning><prediction>I am now
standing in front of the desk. The desk has a lamp, a computer, and several
books on it. The target book is within reach on the right side of the
desk.</prediction></think><answer>rotateright,moveahead,moveahead</answer>
```
User
```
[Initial Observation]:
<image>
Human Instruction: I am looking for a luminous device on my desk to read
books at night. Can you navigate to that object and stay close?
Decide your next action(s).
You can take up to 5 action(s) at a time, separated by ','.
You should first give your thought process with the your observation,
reasoning, and prediction of next state, then your answer.
Both the observation and prediction should describe what you see or expect
to see in the environment.
Your response should be in the format of:
<think><observation>...</observation><reasoning>...</reasoning>
<prediction>...</prediction></think><answer>...</answer>
```
Assistant
```
<think><observation>There is a luminous device on the desk to the right of
the chair. To reach it, I should move forward and then turn
right.</observation><reasoning>Following the strategy, I can go by first
moving forward then turning right to reach the luminous
device.</reasoning><prediction>I will be close to the luminous
device</prediction></think><answer>moveahead, moveahead, moveright</answer>
```
rest of the turns ...

---

 PrimitiveSkill Training Prompt for WorldModeling Strategy 

System
```
You are an AI assistant controlling a Franka Emika robot arm. Your goal is
to understand human instructions and translate them into a sequence of
executable actions for the robot, based on visual input and the instruction.

Action Space Guide
You can command the robot using the following actions:

1. pick(x, y, z) # To grasp an object located at position(x,y,z) in the
robot's workspace.
```

```
2. place(x, y, z) # To place the object currently held by the robot's
gripper at the target position (x,y,z).
3. push(x1, y1, z1, x2, y2, z2) # To push an object from position (x1,y1,z1)
to (x2,y2,z2).

Hints:
1. The coordinates (x, y, z) are in millimeters and are all integers.
2. Please ensure that the coordinates are within the workspace limits.
3. The position is the center of the object, when you place, please consider
the volume of the object. It's always fine to set z much higher when placing
an item.
4. We will provide the object positions to you, but you need to match them
to the object in the image by yourself. You're facing toward the negative
x-axis, and the negative y-axis is to your left, the positive y-axis is to
your right, and the positive z-axis is up.

Examples:
round1:
image1
Human Instruction: Put red cube on green cube and yellow cube on left target
Object positions:
[(62,-55,20),(75,33,20),(-44,100,20),(100,-43,0),(100,43,0)]
Reasoning: I can see from the picture that the red cube is on my left and
green cube is on my right and near me.
Since I'm looking toward the negative x axis, and negative y-axis is to my
left, (62,-55,20) would be the position of the red cube, (75,33,20) would be
the position of the green cube and (-44,100,20) is the position of the
yellow cube.
Also the (100,-43,0) would be the position of the left target, and
(100,43,0) would be the porition of the right target.
I need to pick up red cube first and place it on the green cube, when
placing, I should set z much higher.
Anwer: pick(62,-55,20)|place(75,33,50)
round2:
image2
Human Instruction: Put red cube on green cube and yellow cube on left target
Object positions:
[(75,33,50),(75,33,20),(-44,100,20),(100,-43,0),(100,43,0)]
Reasoning: Now the red cube is on the green cube, so I need to pick up the
yellow cube and place it on the left target.
Anwer: pick(-44,100,20)|place(100,-43,50)

You can take up to 2 action(s) at a time, separated by |.
You should first give your thought process with reasoning and prediction of
next state, and then your answer.
Your response should be in the format of:
<think><observation>...</observation><reasoning>...</reasoning>
<prediction>...</prediction>
</think><answer>...</answer>
e.g. <think><observation>The red cube is on the left at (100,100,40), the
green cube is on the right</observation><reasoning>I need to pick the red
cube at (100,100,40) and place it at
(80,100,60)</reasoning><prediction>After executing this action, the red cube
will be at (80,100,60)</prediction></think
<answer>pick(100,100,40)|place(80,100,60)</answer>
```

User
```
[Initial Observation]:
<image>
Human Instruction: Please put the apple in the drawer and close the drawer.
x_workspace_limit: (-500, -100)
y_workspace_limit: (-600, 200)
z_workspace_limit: (10, 650)
Object positions:
```

```
[(-127, 192, 36), (-500, -537, 200)]
Other information:
{'drawer_open_value': 263}
Decide your next action(s).
You can take up to 2 action(s) at a time, separated by |.
You should first give your thought process with reasoning and prediction of
next state, and then your answer.
Your response should be in the format of:
<think><observation>...</observation><reasoning>...</reasoning>
<prediction>...</prediction></think><answer>...</answer>
```

Assistant

```
<think><observation>The apple is on the ground, right side of the
drawer</observation><reasoning>I need to pick up the apple first, as it is
on the floor. After placing the apple in the drawer, I will close the
drawer.</reasoning><prediction>The apple will be placed in the drawer and
the drawer will be closed.</prediction></think><answer>pick(-127, 192,
36)|place(-500, -537, 500)|push(-500, -537, 500, -500, -537, 650)</answer>
```

rest of the turns ...

---

### SVG Reconstruction Training Prompt for `WorldModeling` Strategy

System
You are a precise SVG code generator.

SVG Quick Guide
Goal: Transform the provided image into precise SVG code that replicates the
image.

Process:
1. First analyze the image carefully, identifying distinct visual elements
2. Identify colors, dimensions, positions, and relationships between elements
3. Generate accurate SVG code that reproduces the image, you cam use path
for better shape

Rewards:
- Overall visual similarity: +5.0
- Structural accuracy: +10.0
Example:
```
<think><observation>I can see an image containing a red circle positioned at
the top-left area of the canvas, approximately at coordinates (25, 25) with
a radius of 15 units. There is also a blue rectangle at the bottom-right
area, sized about 30x20 units and positioned at coordinates (60,
60).</observation><reasoning>Based on my observation, I need to create an
SVG that precisely matches these elements. The circle appears to be slightly
too far right, so I should adjust its x-coordinate to 20 instead of 25. The
rectangle could benefit from being slightly
wider.</reasoning><prediction>After implementing these adjustments, the
resulting SVG should more closely match the original image. I expect the
similarity score to improve to approximately 0.98, as the modified positions
and dimensions will better represent the original
graphic.</prediction></think>
<answer><svg viewBox="0 0 100 100" xmlns="http://www.w3.org/2000/svg">
  <circle cx="20" cy="25" r="15" fill="red" />
  <rect x="60" y="60" width="35" height="20" fill="blue" />
</svg></answer>
```
You can take up to 1 action(s) at a time, separated by ~~.
You should first give your thought process with the your observation and
reasoning, then predict next state, and finally the answer.
Both the observation and prediction should describe what you see or expect
to see in the environment.
Your response should be in the format of:

```
<think><observation>...</observation><reasoning>...</reasoning>
<prediction>...</prediction></think><answer>...</answer>
e.g. <think><observation>I can see an image containing a red circle
positioned at the top-left area of the canvas, approximately at coordinates
(25, 25) with a radius of 15 units. There is also a blue rectangle at the
bottom-right area, sized about 30x20 units and positioned at coordinates
(60, 60).</observation><reasoning>Based on my observation, I need to create
an SVG that precisely matches these elements. The circle appears to be
slightly too far right, so I should adjust its x-coordinate to 20 instead of
25. The rectangle could benefit from being slightly
wider.</reasoning><prediction>After implementing these adjustments, the
resulting SVG should more closely match the original image. I expect the
similarity score to improve to approximately 0.98, as the modified positions
and dimensions will better represent the original
graphic.</prediction></think>
<answer><svg viewBox="0 0 100 100" xmlns="http://www.w3.org/2000/svg">
  <circle cx="20" cy="25" r="15" fill="red" />
  <rect x="60" y="60" width="35" height="20" fill="blue" />
</svg></answer>
User
[Initial Observation]:
<image>
Please carefully observe the image, and generate SVG code that reproduces it
as accurately as possible.
Decide on your SVG code.

You can take up to 1 action(s) at a time, separated by ~~.
You should first give your thought process with the your observation and
reasoning, then predict next state, and finally the answer.
Both the observation and prediction should describe what you see or expect
to see in the environment.
Your response should be in the format of:
<think><observation>...</observation><reasoning>...</reasoning
<prediction>...</prediction></think><answer>...</answer>
Assistant
<think><observation>I can see an image of the letters "kr" inside a circle.
The circle is centered at the top of the image. The letters are inside the
circle, with the circle having a consistent thickness around the
perimeter.</observation><reasoning>The SVG should accurately represent the
circle with a consistent thickness around it and the letters "kr" inside.
The circle should be centered and have a specific size, and the letters
should be placed inside the circle.</reasoning><prediction>The SVG will need
to accurately depict the circle, the letters "kr", and their positioning
within the circle. The circle's size and position should be precise, and the
letters should be correctly placed inside.</prediction></think>
<answer><svg viewBox="0 0 100 100" xmlns="http://www.w3.org/2000/svg">
  <circle cx="50" cy="50" r="25" stroke="black" stroke-width="1" fill="none"
/>
  <text x="50" y="50" font-size="24" font-family="sans-serif"
text-anchor="middle" alignment-baseline="central">kr</text>
</svg></answer>
rest of the turns ...
```

## B.3  Learning Dynamics of RL Baselines

# C  How Can We Represent Internal Beliefs about the World?

In this section, we release the details of different visual state representations. To further understand visual state reasoning, we investigate how different visual state representations affect task performance by prompting models to use the WorldModeling reasoning strategy and requiring them to output specific formats for the <observation> and <prediction> fields.

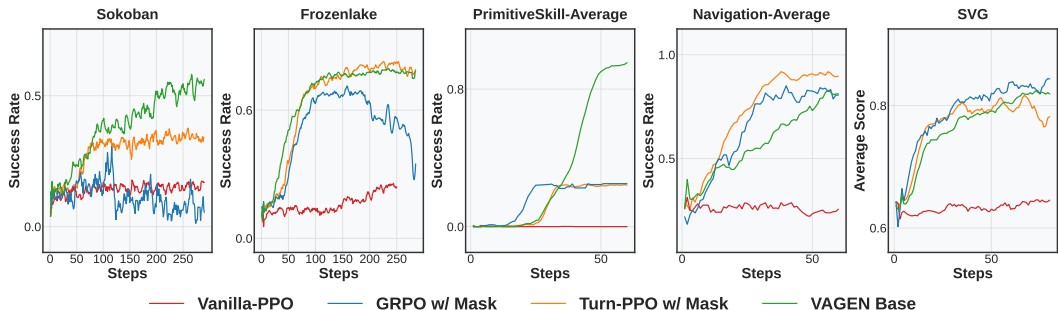

**Figure 7:** Training success rates for RL Baselines and VAGEN-Base

We explore different visual state representations across our environments: Sokoban and FrozenLake support three representation formats (Natural-Lanaguage, Symbolic, and Structured), while PrimitiveSkill supports two formats (Natural-Lanaguage and Structured). For all other tasks, we use Natural-Lanaguage visual state representation as the default. Examples of Natural-Lanaguage visual state representations for all tasks can be found in the prompts provided in Appendix B.2.

The different visual state representation formats are summarized in Table 23:

**Table 23:** Visual state representation formats across different environments.

| Environment | Format | Type | Description |
|---|---|---|---|
| Sokoban | Natural-Lanaguage | Text | Free-form textual descriptions of the game state |
| | Symbolic | Grid | Environment-native grid-based symbols representing game entities |
| | Structured | Dictionary | JSON-like dictionary containing positions of game objects |
| FrozenLake | Natural-Lanaguage | Text | Free-form textual descriptions of the game state |
| | Symbolic | Grid | Environment-native grid-based symbols representing game entities |
| | Structured | Dictionary | JSON-like dictionary containing positions of game objects |
| PrimitiveSkill | Natural-Lanaguage | Text | Free-form textual descriptions of the manipulation state |
| | Structured | Dictionary | JSON-like dictionary containing 3D coordinates of task-relevant objects |

## C.1 Symbolic Representations

For symbolic representations, we use environment-specific symbols to represent different entities in a grid format.

**Sokoban Symbolic Format** uses the following mapping:

```
# Wall | _ Floor | O Target | X Box | P You | * Box on Target | S You on Target
```

An example of the symbolic representation:

```
######
#_P__#
#_XO_#
###__#
######
```

**FrozenLake Symbolic Format** uses the following mapping: _ Frozen | O Hole | G Goal | P Player | X Player fell into hole | * Player on goal

An example of the symbolic representation:

```
___G
O___
__O_
```

_P__

## C.2 Structured Representations

For structured representations, we use dictionary-based formats containing precise positional information of relevant objects.

**Sokoban Structured Format** contains player, box, and target positions along with grid dimensions.

An example of the symbolic representation:

```
{player_position: (4, 2), box_positions: [(3, 2)],
target_positions: [(3, 1)], grid_size: (6, 6)}
```

**FrozenLake Structured Format** contains player, target, and hole positions along with grid dimensions.

An example of the symbolic representation:

```
{player_position: (3, 2), target_position: (3, 2),
hole_positions: [(1, 2)], grid_size: (4, 4)}
```

**PrimitiveSkill Structured Format** contains 3D coordinates of task-relevant objects.

An example of the symbolic representation:

```
{red_cube: (100, 100, 40), green_cube: (200, 200, 60)}
```

These different formats allow us to investigate how the choice of visual state representation affects the model's ability to reason about and predict state transitions across different types of environments.

# D   How to Improve Reasoning over World Models?

## D.1   World Modeling Reward via LLM-as-a-Judge

To enhance visual state reasoning capabilities, we implement an LLM-as-a-Judge framework that evaluates the quality of the agent's `StateEstimation` and `TransitionModeling` reasoning. The judge model used in our experiments is GPT-4.1 nano, which evaluates whether the agent's descriptions and predictions accurately match the ground truth state information.

Our evaluation system uses task-specific prompts for both `StateEstimation` (current state grounding) and `TransitionModeling` (next state prediction) assessment. The prompts are structured to guide the judge model through step-by-step reasoning before providing a binary YES/NO evaluation.

```
┌─ Sokoban StateEstimation Evaluation ─────────────────────┐

  Evaluate whether the description accurately captures the key position
  relationships in the Sokoban game state.
  Answer YES if the directional relationships are correct, or NO if they
  contain directional errors.

  # Context
  You are evaluating whether the description correctly identifies the
  directional relationships between:
  1. The player and the box(es)
  2. The box(es) and the target(s)

  The description doesn't need to be perfectly precise or mention every detail
  - it just needs to have the correct directional relationships (Up, Down,
  Left, Right).

  Example:
```

Groundtruth Current State Information: ['box0 is at the same row and to the left of the player', 'target0 is above and on the left side of the player', 'target0 is above and on the left side of box0']
State Description: The player is below the box and the target is below the box.
- <think>The state description contains spatial relationship information, do further analysis. According to the ground truth data, box0 is at the same row and to the left of the player, target0 is above and on the left side of the player, target0 is above and on the left side of box0. The description states 'The player is below the box and the target is below the box.' The player is actually at the same row as the box (not below), and the target is actually above the box (not below). Both directional relationships are incorrectly identified.</think><answer>NO</answer>

Example:
Groundtruth Current State Information: ['box0 is above and on the right side of the player', 'target0 is above and at the same column as the player', 'target0 is above and on the left side of box0']
State Description: The box is above the player and the target is to the left of the box
- <think>The state description contains spatial relationship information, do further analysis. According to the ground truth data, box0 is above and on the right side of the player, target0 is above and at the same column as the player, target0 is above and on the left side of box0. The description states 'The box is above the player and the target is to the left of the box.' It correctly identifies that the box is above the player (box0 is above and on the right side of the player). It also correctly identifies that the target is to the left of the box (target0 is above and on the left side of box0). Both key directional relationships are accurately described.</think><answer>YES</answer>

# Groundtruth Current State Information:
{state_information_dict}

# State Description:
{natural_language_description}

Think step by step:
  1. Relative Relationship Requirements:
     - Must describe at least one relationships BETWEEN entities (player-box, player-target, box-target)
     - Absolute positions like "player is on the left side" are insufficient
     - Need relational descriptions like "player is left of target"

  2. Essential Relationships to Check
     - Player-Target relationship (highest priority)
     - Player-Box relationship
     - Box-Target relationship

  3. Equivalent Expression Recognition
     - "box is above player" = "player is below box"
     - "target is left of box" = "box is right of target"
     - Must recognize these as identical spatial relationships. Absolute position is not allowed

Your answer should be in the format of
<think>...</think><answer>YES</answer> or
<think>...</think><answer>NO</answer>.

Evaluate whether the prediction correctly anticipates the key position
relationships that will exist in the next Sokoban state.
Answer YES if the predicted directional relationships are correct, or NO if
they contain directional errors.

# Context
You are evaluating whether the prediction correctly identifies the
directional relationships that will exist after the move:
1. The future position of the player relative to the box(es)
2. The future position of the box(es) relative to the target(s)

# Important: The Prediction Comes First
Remember: The Next State Prediction is made BEFORE the Groundtruth Next
State exists. Your task is to check if the prediction correctly anticipated
what actually happened.
If the box and target are at same position, this prediciton is seen as
success immediately (YES)

Example:
Groundtruth Next State Information: ['box0 is above and on the right side of
the player', 'target0 is above and on the left side of the player', 'target0
is above and on the left side of box0']
Next State Prediction: The player will be to the left of the box, and the
box will be to the right of the target.
- <think>The prediction state contains spatial relationship between player
and target, do further analysis. According to the ground truth data, box0 is
above and on the right side of the player, target0 is above and on the left
side of the player, target0 is above and on the left side of box0. The
description states 'The player will be to the left of the box, and the box
will be to the right of the target.' It correctly identifies that the player
is to the left of the box (since box0 is on the right side of the player).
It also correctly identifies that the box is to the right of the target
(since target0 is on the left side of box0). Therefore, this description
correctly identifies the key directional
relationships.</think><answer>YES</answer>

# Groundtruth Next State Information:
{state_information_dict}

# Next State Prediction:
{natural_language_description}

Think step by step:
  1. Relative Relationship Requirements:
     - Must describe at least one relationships BETWEEN entities (player-box,
player-target, box-target)
     - Absolute positions like "player is on the left side" are insufficient
     - Need relational descriptions like "player is left of target"

  2. Essential Relationships to Check
     - Player-Target relationship (highest priority)
     - Player-Box relationship
     - Box-Target relationship

  3. Equivalent Expression Recognition
     - "box is above player" = "player is below box"
     - "target is left of box" = "box is right of target"
     - Must recognize these as identical spatial relationships. Absolute
position is not allowed

Your answer should be in the format of
`<think>...</think><answer>YES</answer>` or
`<think>...</think><answer>NO</answer>`.

## FrozenLake StateEstimation Evaluation

Evaluate whether the description accurately captures the key position
relationships in the FrozenLake game state.
Answer YES if the directional relationships are correct, or NO if there are
errors.

# Context
You are evaluating whether the description correctly identifies:

1. The directional relationship between the player and the goal (MUST Have)
2. The directional relationship between the player and the hole (if present)

The description doesn't need to be perfectly precise - it just needs to have
the correct directional relationships between the player and target (Up,
Down, Left, Right), and between the player and hole if applicable.

# Groundtruth Current State Information:
{state_information_dict}

# State Description:
{natural_language_description}

Think step by step:
1. Player relationship with Goal
   - Goal (Target) MUST include in state description, without target the
description is automatically wrong (NO)
   - If there is no direction between player and goal, like "player is right
to the target", the description is automatically wrong (NO)
   - This takes highest priority over all other considerations

2. Equivalent Expression Recognition
   - "goal is above player" = "player is below goal"
   - "target is left of box" = "box is right of target"
   - Must recognize these as identical spatial relationships. Absolute
position is not allowed

3. Simple Judgment Rule
   - If player at goal -> YES
   - If direction aligns with needed movement -> YES
   - Otherwise -> NO

Your answer should be in the format of
`<think>...</think><answer>YES</answer>` or
`<think>...</think><answer>NO</answer>`.

## FrozenLake TransitionModeling Evaluation

Evaluate whether the prediction correctly anticipates the key aspects of the
next FrozenLake state.
Answer YES if the prediction accounts for directional relationships and
potential holes, or NO if it contains errors.

# Context
You are evaluating whether the prediction correctly identifies:

1. The position relationship between the player and the goal after the prediction

# Important: The Prediction Comes First
Remember: The Next State Prediction is made BEFORE the Groundtruth Next State exists. Your task is to check if the prediction correctly anticipated what actually happened.

The prediction doesn't need to perfectly describe every aspect of the next state - it just needs to correctly anticipate the directional relationships (Up, Down, Left, Right) or address any dangers from holes.

# Groundtruth Next State Information:
{state_information_dict}

# Next State Prediction:
{natural_language_description}

Think step by step:
1. Player relationship with Goal
  - If player is already at the goal position, the prediction is automatically correct (YES)
  - Goal (Target) MUST include in prediction state, without target the prediction is automatically wrong (NO)
  - If there is no direction between player and goal, like "player is right to the target", the prediction is automatically wrong (NO)
  - This takes highest priority over all other considerations

2. Directional Correctness
  - Evaluate if the predicted movement direction aligns with the relative position between player and goal
  - For example, if player is left of goal, moving right is correct
  - **CRITICAL: Recognize equivalent expressions of the same spatial relationship**
     * "player is above target" = "target is below player"
     * "player is left of target" = "target is right of player"
     * These are the SAME relationship expressed from different perspectives

3. Simple Judgment Rule
  - If player at goal -> YES
  - If direction aligns with needed movement -> YES
  - Otherwise -> NO

Your answer should be in the format of
<think>...</think><answer>YES</answer> or
<think>...</think><answer>NO</answer>.

---

Navigation StateEstimation Evaluation

Evaluate whether the description effectively communicates the spatial relationship between the agent and target object, even if the exact directional terms differ.
Answer YES if the overall spatial understanding is correct, or NO if it fundamentally misunderstands the spatial layout.

# Context
You are evaluating whether the description effectively conveys where the target object is located relative to the agent. The exact directional terminology (left, right, ahead, etc.) may differ between the state information and the description, but the important factor is whether the description would lead to correct navigation.

```
# Groundtruth Current State Information:
{state_information_dict}

# State Description:
{natural_language_description}

Think step by step:
1. Check if the description contains spatial relationship between agent and
target object
   - If no spatial relationship is mentioned, answer NO
2. If spatial relationship exists, check if the predicted direction is
consistent with the target direction
   - "ahead/forward" = "ahead"
   - "left" = "left"
   - "right" = "right"
   - Combined directions like "forward-left", "forward-right" are acceptable
if they include the correct primary direction
3. The prediction is correct if it mentions moving toward the target in a
direction that reasonably aligns with the groundtruth direction

Your answer should be in the format of
<think>...</think><answer>YES</answer> or
<think>...</think><answer>NO</answer>.
```

```
Evaluate whether the prediction effectively anticipates how the agent would
navigate toward the target object, even if the exact directional terms
differ.
Answer YES if the overall navigation plan is reasonable, or NO if it
misunderstands or did not mention the spatial layout.

# Context
You are evaluating whether the prediction effectively anticipates how the
agent would move to reach the target object. The exact directional
terminology (left, right, ahead, etc.) may differ between the state
information and the prediction, but the important factor is whether the
prediction would lead to successful navigation.

# Important: The Prediction Comes First
Remember: The Next State Prediction is made BEFORE the Groundtruth Next
State exists. Your task is to check if the prediction correctly anticipated
what actually happened.

# Groundtruth Next State Information:
{state_information_dict}

# Next State Prediction:
{natural_language_description}

Think step by step:
1. First, check if the prediction explicitly uses EXACT directional terms
that appear in the groundtruth state: "ahead", "left", "right", "up", "down".
   - Terms like "move towards", "closer to", "near", "approaching", "in front
of", "by", "at" DO NOT qualify
   - "Will be on the left/right/ahead" or "Will move left/right/forward" DO
qualify
   - If no exact directional match to groundtruth is present, conclude with
NO immediately
```

2. If explicit direction words exist, verify that they EXACTLY match the target object's direction in the groundtruth:
   - If target is "ahead", prediction must specify "ahead", "forward", "slightly left", OR "slightly right" (special case: we allow slightly left/right for ahead targets)
   - If target is "right", prediction must specify "right"
   - If target is "left", prediction must specify "left"
3. Even if the prediction mentions intermediate objects correctly, it MUST explicitly state the correct final direction to the target object
4. The prediction cannot substitute object references for directions (saying "move to X" instead of "move right")
5. Remember that the prediction was made BEFORE the groundtruth state was determined

Your answer should be in the format of
<think>...</think><answer>YES</answer> or
<think>...</think><answer>NO</answer>.

---

PrimitiveSkill StateEstimation Evaluation

Compare the description of the current state with the groundtruth current state information.
Answer YES if the description reasonably matches the current state information, or NO if it doesn't.

# Context
You are evaluating whether an agent's description accurately reflects the actual state. The description should capture the meaningful relationships and positions relevant for decision-making in the task.

Important evaluation criteria:
1. If the description includes coordinates, they don't need to be exact matches with the groundtruth
2. For coordinate values, consider them correct if they are within these error tolerances:
   - For x and y coordinates: within +10 or -10 units of groundtruth
   - For z coordinates: within +10 or -10 units of groundtruth
3. The overall spatial relationships and object identifications should be correct
4. If the description includes a dict-formatted state information, that's good but not required

# Groundtruth Current State Information:
{state_information_dict}

# State Description:
{natural_language_description}

Think step by step:
1. Identify the key objects and their positions in the groundtruth information
2. Check if the description correctly identifies these objects
3. For any coordinates mentioned, check if they are within the acceptable error range
4. Determine if the overall spatial understanding is correct, even if specific numbers differ slightly
5. If coordinates in the description differ from groundtruth but are within the error tolerance, consider them correct

```
Your answer should be in the format of
<think>...</think><answer>YES</answer> or
<think>...</think><answer>NO</answer>.
```

PrimitiveSkill TransitionModeling Evaluation

```
Compare the prediction of the next state with the groundtruth next state
information.
Answer YES if the prediction reasonably matches the next state information,
or NO if it doesn't.

# Context
You are evaluating whether an agent's prediction of the next state is
accurate. The prediction should capture the meaningful changes and
relationships that will result from the planned actions.

# Important: The Prediction Comes First
Remember: The Next State Prediction is made BEFORE the Groundtruth Next
State exists. Your task is to check if the prediction correctly anticipated
what actually happened.

Important evaluation criteria:
1. If the prediction includes coordinates, they don't need to be exact
matches with the groundtruth
2. For coordinate values, consider them correct if they are within these
error tolerances:
   - For x and y coordinates: within +10 or -10 units of groundtruth
   - For z coordinates: within +10 or -10 units of groundtruth
3. The overall predicted movements and resulting spatial relationships
should be correct
4. If the prediction includes a dict-formatted state information, that's
good but not required

# Groundtruth Next State Information:
{state_information_dict}

# Next State Prediction:
{natural_language_description}

Think step by step:
1. Identify the key objects and their positions in the groundtruth next
state information
2. Check if the prediction correctly anticipates these object positions
3. For any coordinates mentioned, check if they are within the acceptable
error range
4. Determine if the overall predicted movement and resulting state is
correct, even if specific numbers differ slightly
5. If coordinates in the prediction differ from groundtruth but are within
the error tolerance, consider them correct

Your answer should be in the format of
<think>...</think><answer>YES</answer> or
<think>...</think><answer>NO</answer>.
```

**Text-based State Information** To provide ground truth information for the judge model, we extract and format state information differently for each environment:

**Sokoban**: We extract the positions of players, boxes, and targets, then convert their relative positions into natural language sentences. For example:

```
['box0 is above and at the same column as the player',
```

```
'target0 is above and at the same column as the player',
'target0 is above and at the same column as box0']
```

**FrozenLake**: Similarly, we convert positional relationships into descriptive sentences:

```
['target is above and at the same column as the player',
 'hole0 is above and at the same column as the player',
 'hole1 is above and on the right side of the player',
 'hole2 is above and on the right side of the player',
 'hole3 is above and on the right side of the player',
 'hole4 is at the same row and to the right of the player']
```

**Navigation**: We use dictionary-based information directly from the environment:

```
{'target_obj_type': 'Toaster',
 'target_distance_to_player': 2.59,
 'target_direction_to_player': 'ahead',
 'visible_objects': [{'type': 'Cabinet',
   'direction_to_player': 'left',
   'distance_to_player': 0.94},
  {'type': 'Drawer',
   'direction_to_player': 'left',
   'distance_to_player': 1.18},
  {'type': 'CounterTop',
   'direction_to_player': 'left',
   'distance_to_player': 1.33},
  ...]}
```

**PrimitiveSkill**: We provide coordinate-based position information:

```
{'lefz_target_position': (80, -100, 0),
 'righz_target_position': (80, 100, 0),
 'red_cube_position': (83, -102, 20),
 'green_cube_position': (-47, 95, 20)}
```

### D.2   Mitigating Reward Hacking via Structured Evaluation and Repetition Penalty

To address reward, we developed and implemented a composite reward mechanism that combines structured evaluation with a penalty for historical repetition.

The method consists of the following key steps:

1. **Natural Language to Structured Reasoning Conversion:** Instead of relying on a holistic, qualitative judgment from the LLM on the agent's natural language reasoning, we first use LLM to parse and convert the reasoning into a **structured dictionary**. This dictionary explicitly captures key entities and relations within the reasoning.

2. **Quantitative Evaluation with F1 Score:** The structured dictionary is then programmatically compared against a ground-truth dictionary representing the optimal reasoning for the given state. By calculating the **F1 score** between these two dictionaries, we obtain a precise and quantitative measure of the correctness and completeness of the agent's reasoning. This approach is significantly more robust than relying on vague text similarity or an LLM's overall impression.

3. **Penalty Mechanism for Low-Entropy and Repetitive Behavior:** We observed that agents can fall into a pattern of over-exploiting previously successful reasoning, repeatedly generating the same high-reward sentences even in inappropriate states. This behavior is reflected in a noticeable decrease in the **entropy** of its outputs. To counteract this, we introduced a penalty mechanism:

   - We maintain a **max-heap** to dynamically track the most frequently generated sentences in the agent's history.
   - When evaluating a new response from the agent, we perform a dual-condition check:

(a) The response sentence matches one of the top frequent sentences in the heap.

(b) Its corresponding F1 score is **below** a predefined correctness threshold.

Only when **both conditions are met** do we conclude that the agent is blindly repeating an incorrect answer and apply a negative penalty.

In our Sokoban experiments, we set the F1 score threshold to $0.7$. If a response was identified as both repetitive and incorrect (F1 $< 0.7$), a penalty of $-0.1$ was applied to the reward. This mechanism forces the agent to explore more diverse and state-relevant reasoning pathways.

## D.3 Bi-Level General Advantage Estimation (GAE) Details and Additional Results

This section provides the detailed pseudo-code for our Bi-Level GAE algorithm, which is shown in Algorithm 2. The key modification from the base RL algorithm is in Phase 2, where we introduce a bi-level advantage estimation process that operates at both turn and token levels. To explore the effectiveness of Bi-level GAE, we also experiment adding Bi-level GAE to Freethink baselines, and results in Table 24 shows that Bi-level GAE is able to serve as a general plug-in strategy for RL training. Results on different model families and sizes are shown in Table 25, training time and token usage are given in 26.

**Table 24:** Bi-level GAE helps `FreeThink` RL baselines, and can serve as a general mechanism in RL training for VLM agents.

| Group / Model | Task | Method | Test Success Rate |
|---|---|---|---|
| Qwen2.5-VL 3B | FrozenLake | FreeThink | 0.39 |
| | FrozenLake | FreeThink + Bi-Level GAE | **0.62** |
| Qwen2.5-VL 3B | Sokoban | FreeThink | 0.43 |
| | Sokoban | FreeThink + Bi-Level GAE | 0.43 |

**Table 25:** Results on different model families and sizes on Sokoban task.

| Group / Model | Method | Test Success Rate |
|---|---|---|
| Qwen2.5-VL 3B | VAGEN-BASE | 0.61 |
| | VAGEN-FULL | 0.79 |
| Qwen2.5-VL 7B | VAGEN-BASE | 0.63 |
| | VAGEN-FULL | **0.92** |
| InternVL3-2B | VAGEN-BASE | 0.36 |
| | VAGEN-FULL | 0.39 |

**Table 26:** Training efficiency for primary experiments (approximate).

| Task | H100 GPU Hours (Approx.) | LLM-as-Judge Token Cost (Approx.) |
|---|---|---|
| FrozenLake | 40 | $2.3 \times 10^7$ |
| Sokoban | 40 | $2.3 \times 10^7$ |
| Navigation | 30 | $6.0 \times 10^6$ |
| ManiSkill | 30 | $4.6 \times 10^6$ |
| SVG | 10 | N/A |

---

**Algorithm 2** Bi-Level General Advantage Estimation (GAE)

---

**Input:**

        Full trajectory token sequence $\bar{\tau}$, organized by turns $t = 0..T-1$.

        Per-turn rewards $r_0, \ldots, r_{T-1}$.

        Critic $V_\phi$, Actor $\pi_\theta$, Reference policy $\pi_{\text{ref}}$.

        Hyperparameters: $\gamma_{\text{turn}}, \lambda_{\text{turn}}, \gamma_{\text{token}}, \lambda_{\text{token}}, \beta$.

**Output:**

        Token-level advantages $A^{\text{token}}_{t,i}$ for each token $i$ in each action $a_t$.

---

**Stage 1: Turn-Level Advantage Estimation**

1: Initialize turn-level advantage $A^{\text{turn}}_T \leftarrow 0$.

2: **for** $t = T-1, \ldots, 0$ (backwards) **do**

3:     Define value at current turn boundary: $V_t \leftarrow V_\phi(\bar{\tau}_{\leq a_t})$.

4:     **if** $t = T-1$ **then**

5:         Define value at next turn boundary: $V_{t+1} \leftarrow 0$.

6:     **else**

7:         Define value at next turn boundary: $V_{t+1} \leftarrow V_\phi(\bar{\tau}_{\leq a_{t+1}})$.

8:     **end if**

9:     Compute turn-level TD-error: $\delta^{\text{turn}}_t \leftarrow r_t + \gamma_{\text{turn}} V_{t+1} - V_t$.

10:     Compute turn-level advantage: $A^{\text{turn}}_t \leftarrow \delta^{\text{turn}}_t + \gamma_{\text{turn}}\lambda_{\text{turn}}A^{\text{turn}}_{t+1}$.

11: **end for**

---

**Stage 2: Token-Level Advantage Estimation**

12: **for** $t = 0, \ldots, T-1$ (forwards through turns) **do**

13:     Let $J = |\mathcal{E}(a_t)|$ be the number of tokens in action $a_t$.

14:     *// Initialize the advantage of the final token of the action with the turn-level advantage.*

15:     Let $\bar{\tau}_{t,J-1}$ be the final token of action $a_t$.

16:     Calculate its KL penalty: $r^{\text{KL}}_{t,J-1} \leftarrow -\beta \cdot \text{KL}\left[\pi_\theta(\cdot|\bar{\tau}_{<(t,J-1)}) \,\|\, \pi_{\text{ref}}(\cdot|\bar{\tau}_{<(t,J-1)})\right]$.

17:     Compute its TD-error: $\delta^{\text{token}}_{t,J-1} \leftarrow r^{\text{KL}}_{t,J-1} + \gamma_{\text{token}}V_\phi(\bar{\tau}_{\leq a_t}) - V_\phi(\bar{\tau}_{<(t,J-1)})$.

18:     Set final token's advantage: $A^{\text{token}}_{t,J-1} \leftarrow \delta^{\text{token}}_{t,J-1} + A^{\text{turn}}_t$.

19:     **for** $i = J-2, \ldots, 0$ (backwards through remaining tokens in the action) **do**

20:         Let $\bar{\tau}_{t,i}$ be the current token.

21:         Calculate KL penalty: $r^{\text{KL}}_{t,i} \leftarrow -\beta \cdot \text{KL}\left[\pi_\theta(\cdot|\bar{\tau}_{<(t,i)}) \,\|\, \pi_{\text{ref}}(\cdot|\bar{\tau}_{<(t,i)})\right]$.

22:         Compute token-level TD-error: $\delta^{\text{token}}_{t,i} \leftarrow r^{\text{KL}}_{t,i} + \gamma_{\text{token}}V_\phi(\bar{\tau}_{<(t,i+1)}) - V_\phi(\bar{\tau}_{<(t,i)})$.

23:         Compute token-level advantage: $A^{\text{token}}_{t,i} \leftarrow \delta^{\text{token}}_{t,i} + \gamma_{\text{token}}\lambda_{\text{token}}A^{\text{token}}_{t,i+1}$.

24:     **end for**

25: **end for**

---

# E   Case Study

In this section, we present a comprehensive case study examining the behavioral patterns and learning dynamics of multi-turn VLM agents across distinct environments. Through systematic analysis of agent responses during different training phases, we investigate how world model reasoning capabilities evolve and identify key phenomena that emerge during the learning process.

### E.1   World Model Reasoning Enhances Spatial Understanding and Multi-Step Planning

Our analysis reveals that incorporating explicit world model reasoning into multi-turn VLM agents significantly improves their spatial understanding and planning capabilities across different environments.

The integration of `StateEstimation` and `TransitionModeling` steps enables agents to better understand relative positions between objects. In Sokoban (Figure 8 (c) and (d)), agents trained with visual reasoning demonstrate improved ability to identify spatial configurations between the player, target, and box. This allows them to effectively navigate around obstacles while maintaining progress toward objectives, contrasting with untrained agents that often make suboptimal moves without proper spatial consideration.

World Model reasoning enables sophisticated multi-step planning. In Navigation tasks (Figure 8 (a) and (b)), agents develop the ability to not only identify target objects and their positions but also recognize blocking objects that impede direct paths. This leads to:

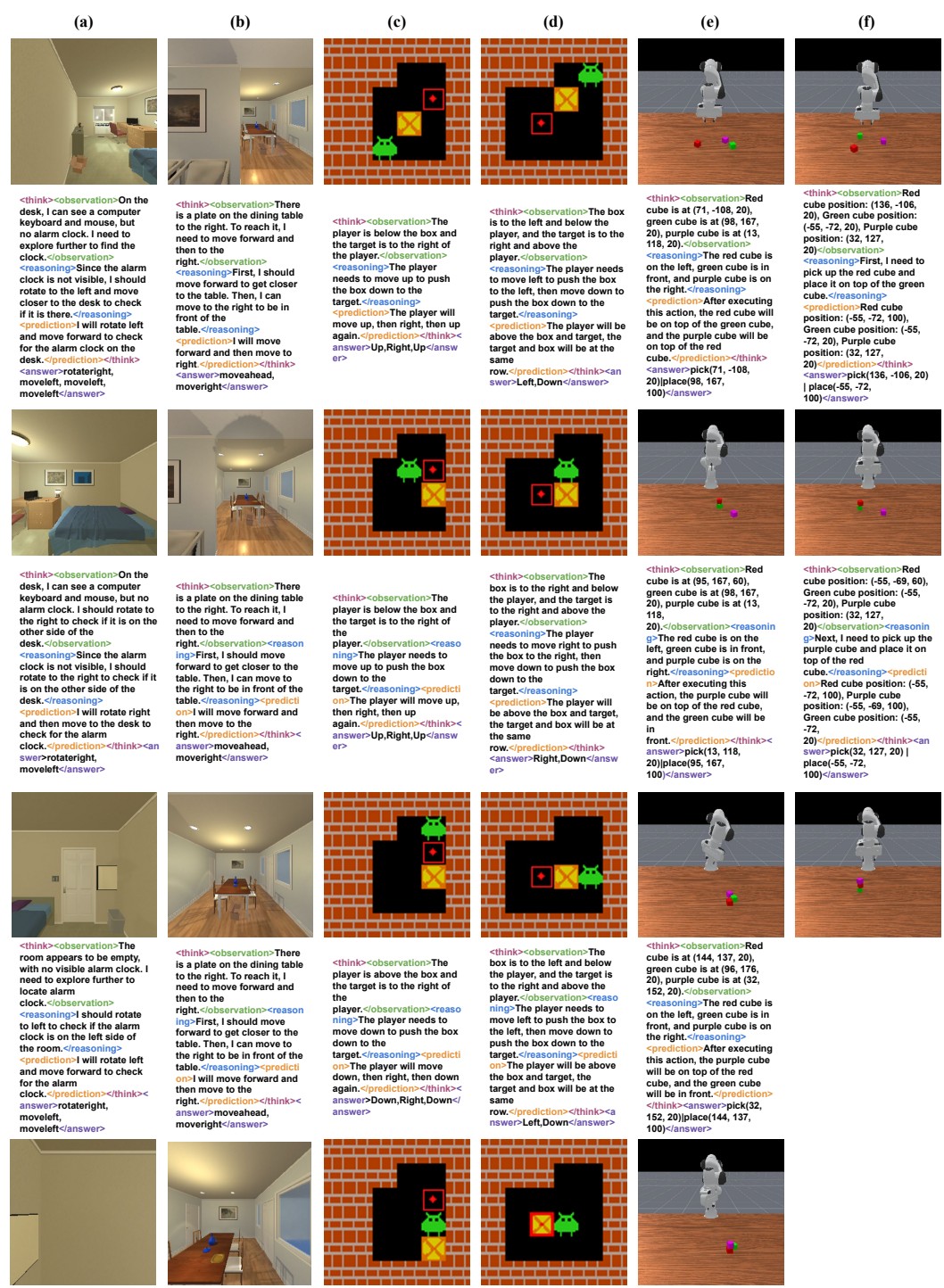

**Figure 8:** Comparison of VAGEN-Base and VAGEN-Full across Navigation, Sokoban, and PrimitiveSkill environments, from left to right. Within each environment, the left column shows VAGEN-Base results while the right column displays VAGEN-Full results. The improvement of VAGEN-Full shows enhanced reasoning ability in understanding environments such as their spatial information.

- **Accurate spatial relationship identification** between targets, obstacles, and agent position

- **Recognition of blocking constraints** that prevent direct paths

- **Multi-step sequence planning** that accounts for both immediate and longer-term goals

## E.2 Response Convergence and Reduced Exploration

We observe a notable pattern of response convergence across different environments, reflecting the fundamental learning dynamics described in the main text. As shown in Cases 1 through 4, agents exhibit dramatically different response behaviors between early and late training phases, with this phenomenon occurring both with and without our `TransitionModeling` Reward as a fundamental aspect of the learning process.

During early training, agents demonstrate strong exploratory tendencies with diverse response patterns. In Sokoban tasks, early responses show varied reasoning approaches, creative problem-solving strategies, and rich descriptive language when analyzing spatial configurations. Similarly, Navigation tasks initially elicit detailed environmental descriptions and multi-faceted planning considerations, reflecting the high entropy characteristic of early-stage learning.

As training progresses, agents develop more streamlined and consistent communication patterns. Late-stage responses become notably more concise and structured, with standardized phrasing that primarily varies in directional or specific action tokens while maintaining clear, efficient expression. This evolution toward templated sentence structures occurs across different environments and reasoning qualities, with entropy steadily decreasing as documented quantitatively in the main analysis.

This convergence represents a fundamental aspect of the learning process, where agents naturally optimize their communication patterns for task effectiveness while exhibiting the reduced exploration behavior characteristic of policy convergence in reinforcement learning.

---

**Sokoban - Early Training**

```
World Modeling - Transition Modeling
State:
['box0 is at the same row and to the right of the player', 'target0 is at
the same row and to the right of the player', 'target0 is at the same place
as box0']
Response:
The player will move down, then to the right, pushing the box to the target.

State:
['box0 is above and at the same column as the player', 'target0 is above and
at the same column as the player', 'target0 is above and at the same column
as box0']
Response:
Push the box down and then move to the right. These actions would bring the
box closer to the target.

World Modeling - State Estimation/Grounding
State:
['box0 is above and at the same column as the player', 'target0 is above and
on the left side of the player', 'target0 is above and on the left side of
box0']
Response:
The player is above the box, and the player is below the target, and there
is a wall between the player and the target. The box is above the player

State:
['box0 is above and on the right side of the player', 'target0 is above and
on the right side of the player', 'target0 is above and on the right side of
box0']
Response:
From the image, the player is at the bottom of the wall, with the box
directly in front of the player to the right and the target directly above
the box.
```

## Sokoban - Late Training

**World Modeling - Transition Modeling**

**State:**
['box0 is at the same row and to the left of the player', 'target0 is above and on the left side of the player', 'target0 is above and at the same column as box0']

**Response:**
The player will move to the left, then push the box left to the target.

**State:**
['box0 is at the same row and to the right of the player', 'target0 is above and on the right side of the player', 'target0 is above and at the same column as box0']

**Response:**
The player will move to the right, then push the box right to the target.

**World Modeling - State Estimation/Grounding**

**State:**
['box0 is above and at the same column as the player', 'target0 is above and on the right side of the player', 'target0 is above and on the right side of box0']

**Response:**
The player is below the box, and the box is below the target.

**State:**
['box0 is below and on the left side of the player', 'target0 is below and on the left side of the player', 'target0 is at the same row and to the left of box0']

**Response:**
The player is to the right of the box, and the box is to the right of the target.

## Navigation - Early Training

**World Modeling - Transition Modeling**

**State:**
```
{'target_obj_type': 'Laptop',
 'target_distance_to_player': 2.58,
 'target_direction_to_player': 'right',
 'visible_objects': [{'type': 'Bed',
   'direction_to_player': 'ahead',
   'distance_to_player': 1.22},
  {'type': 'Pillow',
   'direction_to_player': 'left',
   'distance_to_player': 1.35},
  ...]}
```

**Response:**
I can see a laptop on the desk. I will move forward and right to get closer to the laptop.

**State:**
```
{'target_obj_type': 'Laptop',
 'target_distance_to_player': 2.31,
 'target_direction_to_player': 'ahead',
 'visible_objects': [{'type': 'Drawer',
   'direction_to_player': 'ahead',
   'distance_to_player': 0.85},
  {'type': 'Television',
   'direction_to_player': 'ahead',
   'distance_to_player': 1.3},
  ...]}
```

My next move will be to approach the table where the laptop is placed. The
laptop is placed on the right side of table in front of me.

World Modeling - State Estimation/Grounding
State:
```
{'target_obj_type': 'CellPhone',
 'target_distance_to_player': 2.59,
 'target_direction_to_player': 'ahead',
 'visible_objects': [{'type': 'GarbageCan',
   'direction_to_player': 'ahead',
   'distance_to_player': 2.2},
  {'type': 'CD',
   'direction_to_player': 'ahead',
   'distance_to_player': 2.55},
  ...]}
```
Response:
There is a table with a cell phone on it in the middle of the room. The cell
phone is on the table, close to the center of the room, right beside a trash
can.

State:
```
{'target_obj_type': 'Statue',
 'target_distance_to_player': 2.54,
 'target_direction_to_player': 'ahead',
 'visible_objects': [{'type': 'Drawer',
   'direction_to_player': 'ahead',
   'distance_to_player': 1.99},
  {'type': 'Drawer',
   'direction_to_player': 'ahead',
   'distance_to_player': 1.99},
  ...]}
```
Response:
There is a statue on a table in the room. I need to move forward in order to
reach the statue. The floor is smooth, so I can move forward without
obstacles.

## Navigation - Late Training

World Modeling - Transition Modeling
State:
```
{'target_obj_type': 'DeskLamp',
 'target_distance_to_player': 1.71,
 'target_direction_to_player': 'right',
 'visible_objects': [{'type': 'Chair',
   'direction_to_player': 'ahead',
   'distance_to_player': 1.08},
  {'type': 'Laptop',
   'direction_to_player': 'ahead',
   'distance_to_player': 1.19},
  ...]}
```
Response:
I will move forward-right to reach the lamp.

State:
```
{'target_obj_type': 'Pot',
 'target_distance_to_player': 2.54,
 'target_direction_to_player': 'left',
 'visible_objects': [{'type': 'Bowl',
   'direction_to_player': 'ahead',
   'distance_to_player': 0.88},
```

```
      {'type': 'Potato',
       'direction_to_player': 'ahead',
       'distance_to_player': 1.2},
      ...]}
Response:
I will move forward-left to reach the pot.

World Modeling - State Estimation/Grounding
State:
{'target_obj_type': 'Bowl',
 'target_distance_to_player': 1.52,
 'target_direction_to_player': 'ahead',
 'visible_objects': [{'type': 'Book',
    'direction_to_player': 'right',
    'distance_to_player': 1.18},
   {'type': 'Floor',
    'direction_to_player': 'back',
    'distance_to_player': 1.3},
   ...]}
Response:
There is a round container on the floor. To reach the round container, I
need to move forward.

State:
{'target_obj_type': 'GarbageCan',
 'target_distance_to_player': 1.71,
 'target_direction_to_player': 'ahead',
 'visible_objects': [{'type': 'GarbageBag',
    'direction_to_player': 'ahead',
    'distance_to_player': 1.42},
   {'type': 'GarbageCan',
    'direction_to_player': 'ahead',
    'distance_to_player': 1.71},
   ...]}
Response:
There is a trash can in front of the table to the right. To reach the trash
can, I need to move forward-right.
```

## E.3 Reward Hacking and Over-optimization

We observe interesting patterns of reward optimization behavior, particularly in spatially sensitive environments like FrozenLake where agents learn to adapt their responses to the LLM-as-a-Judge evaluation mechanism. As demonstrated in Cases 5 and 6, agents develop strategic approaches to maximize reward signals, representing a form of reward over-optimization that exemplifies the reward hacking phenomenon described in the main text.

The most notable optimization pattern involves agents converging toward generic, broadly applicable responses that incorporate multiple directional possibilities and general spatial relationships. Rather than providing highly specific, state-dependent analysis, agents learn to produce consistent reasoning templates that reliably satisfy the reward mechanism's evaluation criteria without necessarily reflecting deep, state-specific reasoning. For instance, responses like "The player will be at the position of the gift" emerge as universal patterns that can be applied across diverse state configurations, effectively balancing reward acquisition with computational efficiency while potentially sacrificing reasoning specificity.

This reward hacking behavior becomes more pronounced under Bi-Level training configurations, as highlighted in the main analysis. The enhanced optimization method and turn-aware reward structure enable agents to become more sensitive to reward patterns and develop sophisticated strategies for consistent reward acquisition. The Bi-Level GAE's expanded exploration capabilities and improved credit assignment allow agents to efficiently identify and adapt to the evaluation criteria, demonstrating the system's remarkable effectiveness in learning reward-maximizing behaviors.

However, this optimization may sometimes diverge from the intended reasoning improvements, illustrating the complex dynamics between reward design and genuine capability enhancement in LLM-based evaluation systems.

---

**FrozenLake - Early Training**

World Modeling - Transition Modeling
State:
['target is at the same row and to the left of the player', 'hole0 is at the same place as the player', 'hole1 is below and on the left side of the player', 'hole2 is below and on the right side of the player']
Response:
The player will either go left or right to avoid the hole and continue moving up to reach the icy blocks.

State:
['target is above and on the right side of the player', 'hole0 is above and on the right side of the player']
Response:
The player will be in the bottom center after moving right, and then move down to the center of the gift.

World Modeling - State Estimation/Grounding

State:
['target is below and on the left side of the player', 'hole0 is below and on the left side of the player']
Response:
The player is standing on a block, there is a hole directly in front of the player, and there is a goal at the left corner of the player.

State:
['target is above and on the left side of the player', 'hole0 is above and on the left side of the player', 'hole1 is below and on the left side of the player', 'hole2 is below and at the same column as the player']
Response:
The player is now at the right side of the grid, adjacent to a hole. The goal (G) is on the left side of the player.

---

**FrozenLake - Late Training**

World Modeling - Transition Modeling (Hacking)
State:
['target is at the same place as the player', 'hole0 is below and on the right side of the player']
Response:
The player will be at the position of the gift.

State:
['target is above and on the right side of the player']
Response:
The player will be at the position of the gift.

World Modeling - State Estimation/Grounding (Hacking)
State:
['target is below and on the right side of the player', 'hole0 is above and on the right side of the player', 'hole1 is below and on the right side of the player']
Response:
The player is at the top left, the gift is at the bottom right, and there are holes at the top and bottom right corners.

```
State:
['target is at the same row and to the left of the player']
Response:
The player is in the center of the grid, the gift is on the left side, and
there is a hole on the right side.
```

## F    Extensive Related Work

**RL for LLMs and VLMs.** Recent studies have explored RL for both LLMs and VLMs[80, 70, 58, 20, 79, 65, 40, 12, 14, 13, 24, 71, 74, 77, 16, 75, 59, 78], with approaches ranging from human feedback [80, 70, 58, 65, 20, 40, 71], to rule-based reward functions [12, 13, 74, 24, 77, 14]. For multi-turn RL training, [13] applies PPO to VLMs similar to ours, but we adopt a trajectory-based optimization strategy that better supports POMDP scenarios by leveraging historical context for visual reasoning. Concurrently, [12] and [14] developed hierarchical RL frameworks for LLM alignment in text-only environments. Our work proposes Bi-level GAE, which is similar to the hierarchical design of [12], but with totally different optimization strategy.

**Multi-turn Agent Training for LLMs and VLMs.** Multi-turn interaction is fundamental to agentic tasks for LLMs [12, 67, 63, 14, 57]. For VLMs, this challenge extends to the more complex domain of maintaining consistent visual state representations across interactions [13]. Previous research has explored various approaches including prompting techniques [5, 73, 76] and fixed LLMs/VLMs with additional adapters [72, 68, 60]. Our work extends to a more fundamental level by investigating the integration of multi-round visual state reasoning with reinforcement learning. This approach demonstrates generalizability across diverse benchmarks, including navigation[7], manipulation[69], instruction following[6], image reconstruction [9], and collaborative tasks[66] where VLMs are increasingly utilized as core reasoning engines [74]. Concurrently with our work, several frameworks for training multi-turn VLM agents emerged: GiGPO [61] (verl-agent) targets long-horizon credit assignment, AReaL [62] offers a fully asynchronous, scalable RL system for large reasoning/agentic models, and DART [64] improves GUI agents via decoupled training and adaptive data curation. Our work focus on building internal world models through explicit visual state reasoning in multi-turn VLM agent training.

## G    Summary of Findings

Our investigation into reinforcing visual state reasoning for multi-turn VLM agents, facilitated by the VAGEN training framework, has yielded several key findings regarding the efficacy of explicit reasoning, optimal state representations, specialized RL techniques, and observed agent behaviors during training. These insights are crucial for developing more robust and capable VLM agents that can understand and interact with dynamic visual environments. Below is the summary of our findings.

> **Finding 1: Explicit Visual State Reasoning is Crucial for Multi-Turn VLM Agents**
>
> Vanilla VLMs struggle with multi-turn agentic tasks requiring visual state understanding. Integrating explicit visual state reasoning steps—specifically `StateEstimation` and `TransitionModeling` —into the VLM's thinking process during RL training significantly enhances task performance. The combined `WorldModeling` strategy, in particular, demonstrates strong and stable performance, enabling a trained open-source VLM to outperform its un-trained counterpart and even surpass benchmarked proprietary models.

**Finding 2: Optimal Visual State Representation is Task-Dependent**

The choice of representation for visual states during explicit reasoning significantly impacts performance.

- **Natural Language:** Performs consistently well, especially when structured information must be inferred from raw visual input.

- **Structured Formats:** Excel in manipulation-heavy tasks (e.g., PrimitiveSkill) where object-centric state abstractions are readily available.

- **Symbolic Representations:** Proved less effective due to the model's limited prior interpretability from visual input.

**Finding 3: RL with World Modeling Rewards and Bi-Level GAE Enhances Reasoning Quality and Task Success**

To specifically improve visual state reasoning, VAGEN incorporates:

- **Turn-level World Modeling Reward:** An LLM-as-a-Judge assesses the accuracy of the VLM's explicit state descriptions and predictions, effectively supervising reasoning.

- **Bi-Level General Advantage Estimation (GAE):** Estimates advantages at both turn and token levels, providing finer-grained reward signals and improving credit assignment.

This approach consistently outperforms Base RL, leading to improved reasoning quality, higher task success rates, and better generalization.

### Finding 4: Emergent Reasoning Patterns and Challenges

Beyond quantitative measurements, we qualitatively analyzed how agents learn to reason:

- **Reasoning Stability Varies by Task:** While reasoning in tasks like Navigation and PrimitiveSkill (and often Sokoban) appears relatively normal and beneficial with explicit rewards, tasks like FrozenLake show more erratic reasoning patterns, potentially correlating with its lower performance and the difficulty of its visual state reasoning.

- **Potential for Reward Hacking:** Instances of "reward hacking" were observed, particularly with certain reward configurations (e.g., Bi-Level world modeling in some contexts). Agents might learn to generate reasoning-like text that satisfies the reward mechanism (e.g., format reward, or even LLM-judge for simple cases) without genuinely reflecting deep understanding or accurate future prediction. This suggests the LLM-as-a-Judge mechanism, while an improvement, is not infallible and can be gamed.

- **Bi-Level GAE as a Double-Edged Sword:** While Bi-Level GAE can improve credit assignment, its interaction with World Modeling Rewards might sometimes allow for more "divergent" or less grounded thinking if the reasoning reward itself can be easily hacked. Well-designed, hard-to-game reasoning rewards are crucial for Bi-Level GAE to be consistently beneficial; otherwise, it might amplify the effects of low-quality process-reward.

- **Convergence to Standardized Phrasing:** Regardless of initial hacking or reasoning quality, agents across different environments tend to converge towards using a more uniform, templated sentence structure for their reasoning and actions over prolonged training, which varies only in some action tokens. And we also observed a huge decrease of entropy during training. This pattern may reflect a learned efficiency for tasks with discrete action spaces, but it may also signal reduced diversity or creativity in reasoning.

- **Rule-Based Filtering as a Potential Mitigation:** For simpler forms of reward hacking where reasoning outputs fail basic semantic checks (e.g., not mentioning valid actions in a grid world), simple rule-based filtering before reward assignment could be a pragmatic interim solution.

These observations underscore that while explicit reasoning and rewards are beneficial, the design of these rewards must be robust against exploitation, and continuous monitoring of reasoning quality is essential.

