# OpenReview forum: "VAGEN: Reinforcing World Model Reasoning for Multi-Turn VLM Agents"
_NeurIPS.cc/2025/Conference — NeurIPS 2025 poster_

### Official Review · Reviewer_ortS · 2025-06-28

**Clarity:** 3
**Significance:** 3
**Originality:** 2
**Rating:** 4
**Confidence:** 3

**Summary:**

This paper explores how to improve the performance of Vision-Language Models (VLMs) on multi-turn agent tasks by augmenting reinforcement learning (RL) with explicit visual state reasoning. The authors introduce reasoning strategies such as Grounding and WorldModeling, and propose a new RL framework that integrates (1) a Visual Reasoning Reward and (2) a Bi-Level Generalized Advantage Estimation (GAE) mechanism to more effectively guide the learning process.

**Questions:**

- For Figure 3, would it be possible to show longer training curves? RL training is often noisy, and it is unclear whether the final decline in Base RL is due to temporary fluctuations or actual performance degradation. Also, does Table 3 report results after convergence, or at a selected checkpoint?

**Ethical Concerns:**

["NO or VERY MINOR ethics concerns only"]

**Final Justification:**

The authors report additional experiments on the weaknesses, therefore I will keep my original positive score.

**Limitations:**

Yes.

**Quality:**

3

**Strengths And Weaknesses:**

### Strengths
- The paper is well-written and easy to follow.
- The topic is interesting and relevant to the growing interest in multimodal agent training.
- The experiments span a wide range of environments with different task structures and action space.

Weaknesses
- In the section “Forcing Off-the-Shelf VLMs to Reason Cannot Solve Multi-turn Agentic Tasks”, the authors conclude that most models don't perform well when being prompted to reason. However, many of the evaluated models were not explicitly trained for reasoning (except for Claude 3.7 Sonnet). Given this, poor performance may be expected because the tasks involve spatial reasoning and planning. It would strengthen the claim if the authors included stronger recent models such as Qwen3 series or GPT o3/o4-mini in the benchmark.
- In the PrimitiveSkill task, it appears that Base RL performs better or comparably to Visual Reasoning RL in terms of sample efficiency and final performance where the plots are truncated. The paper states that “Visual Reasoning RL consistently outperforms Base RL,” but this may need to be more carefully qualified. Can the authors provide further explanation? Is there variance across seeds or task subtasks?
- Since Base RL already includes Grounding + WorldModeling, the contrast between Base RL and Visual Reasoning RL mainly lies in reward shaping and advantage estimation. However, more baselines such as direct action-level RL would be helpful to isolate the contribution of reasoning components.
- Since RL methods are often sensitive to random seeds, it would be better to report variance across multiple runs.

---

> ### Author Rebuttal · Authors · 2025-07-31
>
> We sincerely thank Reviewer ortS for the detailed and constructive feedback. We are encouraged that the reviewer finds our paper to be **well-written** and **easy to follow**, with an **interesting and relevant topic** and **experiments that span a wide range of environments**.
>
> The reviewer's main concerns are related to the strength of our baselines, the performance analysis on the PrimitiveSkill task, the need for additional RL comparisons, and the reporting of variance. We believe our clarifications and the new experimental results provided below will address these points.
>
> ---
>
> ### **1. (Regarding Weakness 1) Strength of Baseline Models**
>
> **TL;DR**: We added experiments with stronger proprietary models (OpenAI GPT-o3 and GPT-o4-mini). Our 3B VLM outperforms these models on most tasks, demonstrating the effectiveness of our method.
>
> We appreciate the reviewer’s suggestion to benchmark against stronger recent models. To address this, we ran additional experiments.
>
> We compared our method against strong proprietary models like GPT-o3 and GPT-o4-mini. It is important to note that these models were used with a 4096-token budget (much larger than the 500-token budget for our method and other baselines). We also trained the model for longer steps in Sokoban and FrozenLake using our method (300 vs. 150 in the original submission).
>
> | Model | FrozenLake | Sokoban | Navigation | Primitive Skill | SVG | **Overall** |
> | :--- | :---: | :---: | :---: | :---: | :---: | :---: |
> | GPT-o3 | 0.78 | 0.60 | 0.81 | 0.63 | 0.75 | 0.71 |
> | GPT-o4-mini | 0.82 | 0.44 | 0.75 | 0.50 | 0.78 | 0.66 |
> | **Ours**  | 0.74 | **0.79** | **0.82** | **0.94** | **0.79** | **0.82** |
>
> Our 3B VLM outperforms both GPT-o3 and GPT-o4-mini on most tasks, highlighting the effectiveness of our method.
> *(Note: Qwen3 only supports text inputs. We will test it when its vision-language version releases.)*
>
> ### **2. (Regarding Weakness 2) Clarification on the PrimitiveSkill Task Performance**
>
> **TL;DR**: We acknowledge the reviewer's observation is accurate. Our method's main benefit in the PrimitiveSkill task is improved **sample efficiency in early training**, while performance converges to Base RL in later stages due to task-specific properties. We will revise our claims to reflect this nuance.
>
> We thank the reviewer for this insightful observation. We agree that our original statement "Visual Reasoning RL consistently outperforms Base RL" needs to be more carefully qualified for the PrimitiveSkill task.
>
> *   We re-analyzed PrimitiveSkill by averaging results across its subtasks.
>
> | Step | Base RL| Visual Reasoning RL|
> |:--|:--|:--|
> | 0    | 0.000 | 0.000|
> | 5    | 0.000 | 0.000|
> | 10   | 0.000 | 0.007|
> | 15   | 0.000 | 0.062|
> | 20   | 0.019 | 0.311|
> | 25   | 0.061 | 0.495|
> | 30   | 0.188 | 0.717|
> | 35   | 0.283 | 0.835|
> | 40   | 0.541 | 0.887|
> | 45   | 0.842 | 0.932|
> | 50   | 0.927 | 0.993|
> | 55   | 0.925 | 0.952|
> | 60   | 0.976 | 0.969|
>
>
>
> 1.  The averaged results show Visual Reasoning RL learns faster and achieves higher performance in the initial training phases, confirming its ability to improve training efficiency.
> 2.  In the later stage, Visual Reasoning RL converges to Base RL. We attribute this to two reasons: (i) The PrimitiveSkill task provides object coordinates directly, reducing the need for visual grounding, which is a key strength of our reasoning component. (ii) The action space is relatively predictable, allowing the base agent to eventually learn the world model without extra supervision.
>
> *   **Edits to the paper:** We will edit our manuscript to state that Visual Reasoning RL provides significant sample efficiency improvements, especially in early training, while acknowledging that performance may converge in later stages for tasks with reduced grounding complexity. Thank you for helping us present our contributions more precisely.
>
> ### **3. (Regarding Weakness 3) Isolating Contributions with More RL Baselines**
>
>
> **TL;DR:** Our paper already includes an action-level RL baseline (“No-Think RL”, Table 1), which corresponds to the model directly predicting actions without reasoning. We further added new baselines (Vanilla-PPO, GRPO w/mask, Turn-PPO w/mask). Our method outperforms the next-best baseline by **49%**, highlighting the effectiveness of Visual Reasoning RL.
>
> We thank the reviewer for this suggestion. Our paper (Table 1) already includes an action-level RL baseline (“No-Think RL”), which removes the reasoning step entirely and lets the model output actions directly. This setting serves as a strong reference point for RL without explicit reasoning.
>
> To further isolate the effect of our proposed components (visual reasoning reward and Bi-Level GAE), we conducted additional experiments with the following baselines:
>
> * **Vanilla-PPO:** Standard PPO without state masking.
> * **GRPO (w/mask):** Trajectory-level GRPO with state masking.
> * **Turn-PPO (w/mask):** PPO variant where all tokens in one turn share the same advantage.
>
> | Method            | Overall Performance |
> | ----------------- | :-----------------: |
> | Vanilla-PPO       |         0.26        |
> | GRPO (w/mask)     |         0.54        |
> | Turn-PPO (w/mask) |         0.55        |
> | **Ours**          |       **0.82**      |
>
> Our method outperforms the next-best baseline by **49%**, confirming that our method is effective.
>
>
> ### **4. (Regarding Weakness 4) Reporting Variance Across Multiple Runs**
>
> **TL;DR**: We have conducted new experiments on FrozenLake and Sokoban with random seeds. The results show that our Visual Reasoning RL achieves a higher mean performance.
>
> We agree that reporting variance is crucial for RL results.
>
> *   We have run 3 experiments with random seeds for both Base RL and Visual Reasoning RL on the FrozenLake and Sokoban tasks. We report the mean and variance below.
>
> | Method | FrozenLake | Sokoban |
> |--------|------------|---------|
> | Base RL | 0.7000 ± 0.0557 | 0.6233 ± 0.0808 |
> | Visual Reasoning RL | 0.7267 ± 0.0115 | 0.7800 ± 0.0854 |
>
> *(Note: Results shown as mean ± standard deviation)*
>
>
> The results confirm that Visual Reasoning RL performs better on average.
>
> *  Due to computational constraints, we cannot feasibly run multiple seeds across all ~40 experiments (including tasks, strategies, and ablations). Each full run requires 10~40 H100 GPU hours, making multiple seeds for every setting prohibitive.
>
> ### **5. (Regarding Question 1) Longer Training Curves and Table 3 Reporting**
>
> **TL;DR**: We provide longer training curves for FrozenLake and Sokoban, which show the previous decline in Base RL was a temporary fluctuation. Table 3 reports the best checkpoint for fair comparison. Our new multi-run results confirm that Visual Reasoning RL still achieves superior average performance despite this.
>
> Thank you for this question. RL training can indeed be noisy.
>
> *   **Longer Training Curves:** We have extended the training for FrozenLake and Sokoban to 300 steps.
>
> | Step | FrozenLake Base RL |  FrozenLake Visual Reasoning RL | Sokoban Base RL | Sokoban Visual Reasoning RL |
> | :-- | :--- | :--- | :--- | :--- |
> | 1 | 0.125 | 0.140625 | 0.0390625 | 0.0546875 |
> | 20 | 0.21875 | 0.15625 | 0.125 | 0.125 |
> | 40 | 0.453125 | 0.4296875 | 0.25 | 0.2734375 |
> | 60 | 0.5390625 | 0.5703125 | 0.265625 | 0.390625 |
> | 80 | 0.6953125 | 0.59375 | 0.3515625 | 0.4375 |
> | 100 | 0.734375 | 0.65625 | 0.3515625 | 0.5390625 |
> | 120 | 0.7578125 | 0.71875 | 0.3828125 | 0.5 |
> | 140 | 0.78125 | 0.796875 | 0.4140625 | 0.5546875 |
> | 160 | 0.75 | 0.765625 | 0.421875 | 0.7265625 |
> | 180 | 0.78125 | 0.765625 | 0.4375 | 0.6796875 |
> | 200 | 0.7578125 | 0.7890625 | 0.5078125 | 0.75 |
> | 220 | 0.7890625 | 0.71875 | 0.5703125 | 0.71875 |
> | 240 | 0.78125 | 0.7578125 | 0.5546875 | 0.78125 |
> | 260 | 0.78125 | 0.8203125 | 0.5234375 | 0.6640625 |
> | 280 | 0.78125 | 0.84375 | 0.5546875 | 0.7265625 |
>
>
> The longer curves show that the decline in Base RL was a temporary fluctuation. However, our Visual Reasoning RL still achieves higher final performance.
>
> *   **Clarification on Table 3:** For Table 3, we reported results from the best-performing checkpoint for all methods to ensure a fair comparison. We will clarify this reporting methodology in the paper.

---

> > ### Comment · Reviewer_ortS · 2025-08-04
> >
> > Thank you for the additional experiments. I would like to keep my original positive score.

---

> ### Author Response · Authors · 2025-08-05
>
> Dear Reviewer ortS,
>
> We sincerely thank you for maintaining your positive score and for your constructive feedback throughout the review process. Your comments and suggestions were highly insightful and have greatly helped us strengthen our paper. If you have any remaining unaddressed concerns, please feel free to let us know, and we would be happy to provide further clarification. Your guidance has been invaluable, and we deeply appreciate your engagement in improving our work.
>
> Best,
>
> Authors of Submission 19716

---

### Official Review · Reviewer_wZ7m · 2025-07-01

**Clarity:** 3
**Significance:** 3
**Originality:** 2
**Rating:** 5
**Confidence:** 3

**Summary:**

The authors analyze VLMs in multi-turn reasoning tasks. In particular, they cast the problem as a POMDP, where agents (policies) are trained using RL in partially observed domains.
They test five reasoning strategies, which correspond to different elements that must be taken into consideration during the interactions with the environment, and three visual state representations. They also try to reinforce directly the quality of visual state reasoning, by proposing BI-Level GAE, a General Advantage Estimation algorithm (that propagates advantages at the episode level as well) which is better suited for this. The tests show better success than the same VLMs without training, even though some strong proprietary model are still very competitive.

**Questions:**

Why did you use BI-Level GAE only to train visual state representations, and not for the reasoning in the first part? It seems pretty general and an improvement over base RL, so why not testing it?

Why do you state, in the caption of Table 1, that the proprietary models are non-comparable? Because they do not apply your proposed reasoning strategy?

**Ethical Concerns:**

["NO or VERY MINOR ethics concerns only"]

**Final Justification:**

The authors addressed my main complaints so I raised the score.

**Limitations:**

Not really discussed, only briefly

**Quality:**

3

**Strengths And Weaknesses:**

Strengths

(Quality)
The evaluation is extensive, analyzing the problem along various axes, such as reasoning strategies and visual representations

(Originality and significance) Casting the multi-turn problem as a POMDP is sound and interesting, especially in a field where, often, RL is conflated with 'contextual bandits'. This formalization makes clear that the action of the agent has an effect on the environment. However, it is definitely not groundbreaking, Moreover, Bi-Level GAE seems to be an improvement over base GAE, but its impact on the field or novelty is not clear to me.

(Clarity) It a well-written paper.

Weaknessess

(Quality)
It would be preferrable to compare, like in Table 1, reasoning with BI-Level GAE vs base GAE and non-reasoning (see my questions)

(Clarity)
I believe after line 256 an equation is missing, an important one too.
In general, the introduction could be improved, especially in the end. In particular, at a first read it is not immediately clear how the last contribution differs than the first ones. Also, a clear, 'explicit' summary of the contributions is preffered at the end of the section.

(Quality)
I feel the SVG task is a bit worse than the other environments: it does not fit that well with the multi-turn concept

In summary, I like the paper and the contributions. However, it falls short of a stronger accept due to its potential impact and novelty which are difficult for me, at this moment, to rate too highly. For example, I would give a higher score if the obtained results were overwhelmingly better than the ones obtained by most baselines, even proprietary/advanced models (greater impact), or/and if the new reasoning strategy and training technique were to be compared to other SOTA strategies, showing improvements across the board and producing greater insights (novelty).

---

> ### Author Rebuttal · Authors · 2025-07-31
>
> We sincerely thank Reviewer wZ7m for the detailed and constructive feedback. We are encouraged that the reviewer finds our work to be a **well-written paper** with **extensive evaluation**, and considers our POMDP formalization of the multi-turn problem as **sound and interesting**.
>
> The reviewer’s main concerns seem to arise from the clarity of our experimental design and the significance of the performance gains achieved by our method. We believe our clarifications and the new experimental results provided below can resolve these concerns.
>
> ---
>
> ### **1. (Regarding Weakness 1 & Question 1) Clarification on the experiment design and new Bi-Level GAE experiments.**
>
> **TL;DR**: Bi-Level GAE was designed for fine-grained credit assignment when turn-level visual reasoning rewards are introduced. New experiments confirm that **Bi-Level GAE also improves performance with Free-Think**, but its full benefit emerges under fingrained intermediate rewards.
>
>
> *   **Why we didn't use Bi-Level GAE in the first part**:
>     In the first part, each task’s reward design was **coarse-grained**, and our focus was to study the impact of reasoning strategies rather than the RL algorithm itself. Most tasks only provide (i) a simple turn-level format reward, which is easy to learn per our experimental observations, and (ii) a trajectory-level task-success reward. Therefore, we adopted a standard RL algorithm (with modifications for multi-turn tasks) instead of introducing Bi-Level GAE at this stage.
>
> *  **Why Bi-Level GAE was introduced later**:
> When we incorporated turn-level visual reasoning rewards, we observed that fine-grained credit assignment became crucial. The standard advantage estimation failed to effectively propagate supervision across turns. This motivated the development of Bi-Level GAE, which explicitly combines turn-level and trajectory-level credit assignment to better leverage intermediate reasoning signals.
>
> *   **New Experimental Results:** We appreciate the reviewer’s suggestion to test Bi-Level GAE more broadly. We have now run new experiments applying Bi-Level GAE directly to the `FreeThink` strategy.
>
> | Method | FrozenLake | Sokoban |
> | :--- | :---: | :---: |
> | FreeThink | 0.39 | 0.43 |
> | FreeThink + Bi-Level GAE | **0.62** | **0.43** |
>
> While the effect on Sokoban is small, FrozenLake shows a substantial gain, validating the effectiveness of Bi-Level GAE. We hypothesize that the degree of improvement varies because Bi-Level GAE is most effective when it can propagate **correct and strong intermediate reward signals**. In our full method, the visual reasoning reward provides exactly this kind of dense, turn-by-turn signal. However, in the base task setup, the intermediate rewards are often sparse, with the most significant reward only being delivered upon final task success. This sparsity can limit Bi-Level GAE from demonstrating its full advantage.
>
> ### **2. (Regarding  Weakness 2) Edits for missing equation and contribution summary.**
>
> **TL;DR** We thank the reviewer for pointing out these clarity issues. We will add the missing equations and a more explicit, structured summary of contributions to the camera-ready version.
>
> *   We will add the formal definitions for the turn-level delta ($\delta_n^{\text{turn}}$) and advantage ($A_n^{\text{turn}}$) after line 256:
>
>     $$
>     \delta_n^{\text{turn}} = r_n + \gamma_{\text{turn}} V_\phi(p_{n+1}) - V_\phi(p_{n})
>     $$
>     $$
>     A_n^{\text{turn}} = \delta_n^{\text{turn}} + \gamma_{\text{turn}} \lambda^{\text{turn}} A_{n+1}^{\text{turn}}
>     $$
>
> *   We will revise the end of the introduction as follows:
>     > Our contributions are fourfold. (1) We provide empirical evidence for the importance of explicit visual state reasoning in multi-turn RL for VLM agents. (2) We analyze how different visual state representations affect reasoning quality and task performance. (3) We introduce visual reasoning rewards to guide visual state reasoning and Bi-Level GAE for fine-grained credit assignment. (4) We present VAGEN, a scalable training framework that decouples environment setup from model training, enabling efficient experimentation and algorithmic extensibility. Together, these contributions establish a principled pathway for developing VLM agents capable of robust visual reasoning in dynamic environments.
>
> *  We will also add a brief summary of contributions at the beginning of each relevant section to improve readability.
>
> ### **3. (Regarding  Weakness 3) Clarification on the inclusion of the SVG task.**
>
> We thank the reviewer for raising this point. We included SVG:
> (1) to evaluate whether our method extends beyond tasks with well-defined success and limited action space. SVG introduces **open-ended text generation** and **non-binary evaluation metrics** (DreamSim, DINO);
> (2) we reformulated this task from single-turn to multi-turn, inspired by findings that converting single-turn problems into multi-turn dialogues with feedback can improve LLM reasoning [Liu et al., 2025; Zheng et al., 2025; Shinn et al., 2023]. Our work represents an early but important step in exploring this in the VLM domain.
>
> *Reference:*
>
> *Liu et al., 2025. "A Simple 'Try Again' Can Elicit Multi-Turn LLM Reasoning."*
>
> *Zheng et al., 2025. "What Makes Large Language Models Reason in (Multi-Turn) Code Generation?"*
>
> *Shinn et al., 2023. "Reflexion: Language Agents with Verbal Reinforcement Learning."*
>
>
> ### **4. (Regarding Summary/Impact) New results demonstrating significant performance gains.**
>
> **TL;DR** We have conducted **longer training and new baseline comparisons**, which show that our method achieves **overwhelmingly better** results. Our lightweight **3B VLM** now **outperforms best evaluated proprietary model by 14% overall** and **surpasses other RL baselines by 49%**, demonstrating clear significance and impact.
>
> * **New Results vs. Advanced Models:** We trained our model for longer in FrozenLake and Sokoban (300 steps vs. 150) and added comparisons against Open AI o3 and Open AI o4‑mini. The results show our 3B parameter model now achieves a new state-of-the-art.
>
> | Model                 | FrozenLake |  Sokoban | Navigation | Primitive Skill |  SVG | **Overall** |
> | :-------------------- | :--------: | :------: | :--------: | :-------------: | :--: | :---------: |
> | Claude 3.7            |    0.69    |   0.25   |    0.56    |       0.28      | 0.88 |     0.52    |
> | GPT‑4o                |    0.54    |   0.43   |    0.72   |       0.50      | 0.81 |     0.60    |
> | Open AI o4‑mini |    **0.82**    |   0.44   |    0.75    |       0.75      | 0.56 |     0.67    |
> | Open AI o3                    |    0.78    |   0.60   |    0.81    |       0.75      | 0.66 |     0.72    |
> | **Ours (Qwen‑VL 3B)** |  0.74  | **0.79** |  **0.82**  |     **0.94**    | 0.79 |   **0.82**  |\
>
> Our relatively small 3B model outperforms Open AI o3 by **+14% overall**.
>
> * **New Results vs. RL Baselines:** To better highlight the novelty and contribution of our method, we also ran new experiments comparing against other RL baselines.
>
> | Method            | Overall Performance |
> | :---------------- | :-----------------: |
> | Vanilla‑PPO       |         0.26        |
> | GRPO (w/mask)     |         0.54        |
> | Turn‑PPO (w/mask) |         0.55        |
> | **Ours**          |       **0.82**      |
>
> Our method leads the next‑best baseline by **+49%**.
>
> * We believe the two new experiments above confirm the effectiveness and significance of our method.
>
>
> ### **5. (Regarding Question 2) Clarification on "non-comparable" models.**
>
> We apologize for the confusing phrasing. "Non-comparable" was intended to mean "not a controlled comparison". Our intent was to emphasize that for a scientifically rigorous evaluation of **reasoning strategies** and **optimization algorithms**, it is crucial to hold the base model constant (i.e., same size and architecture). We will correct the phrasing in the camera-ready version to "not a controlled comparison" to avoid future misunderstanding.

---

> > ### Comment · Reviewer_wZ7m · 2025-08-03
> > **Response to authors**
> >
> > I thank the authors for their extensive response, which addressed my main complaints satisfactorily. I urge them to improve the final paper by adding the clarification about the SVG domain and all the other details/improved results discussed in their rebuttal.

---

> ### Author Response · Authors · 2025-08-05
>
> Dear Reviewer wZ7m,
>
> We are pleased to know that our rebuttal satisfactorily addressed your concerns. We sincerely thank you for your invaluable points regarding Bi-Level GAE, paper edits and clarifications, the role of the SVG task, and other helpful suggestions. We will incorporate the expanded experiments, clearer equations, a concise contribution recap, and the detailed SVG discussion, as well as all updated results, into the paper to ensure that your insightful feedback is fully reflected.
>
> Best,
>
> Authors of Submission 19716

---

### Official Review · Reviewer_77zp · 2025-07-02

**Clarity:** 3
**Significance:** 3
**Originality:** 3
**Rating:** 5
**Confidence:** 3

**Summary:**

The paper tackles the challenge of training VLMs to act as effective agents in multi-turn tasks like Sokoban agme. The obstacle as the authors mention is the VLM's difficulty in reasoning about changing visual states over time. To address this problem, the paper proposes "Visual Reasoning RL". They show that making the VLM explicitly reasoning about its environment significantly boosts performance, inluding grounding and worldmodeling. The paper also explores the way to represent the visual states. It finds that natural language is effective for most tasks. There are two techniques introduced to specifically improve the quality of visual reasoning (a turn-level visual reasoning award, and Bi-level general advantage estimation). The experiments demonstrate that this approach could improve task success rates and the quality of the reasoning.

**Questions:**

The questions are in the section "Strengths And Weaknesses"

**Ethical Concerns:**

["NO or VERY MINOR ethics concerns only"]

**Final Justification:**

My concerns have been addressed. I'll maintain the current score (5).

**Limitations:**

yes

**Quality:**

4

**Strengths And Weaknesses:**

## Strengths
1. The work is complementary to existing RL-trained reasoning VLMs. It extends the current one-turn reasoning VLMs (usually tackling math problems) to support multi-turn reasoning tasks like Sokoban.
2. The experiment is conducted on many environments/tasks, such as Sokoban, FrozenLake, Navigation, ... The ablation shows the effectiveness of VLM doing reasoning, the grounding and the worldmodeling. it also compares with a few baselines such as Qwen2.5-VL and close source models (e.g., GPT-4o), showing that the proposed framework is performant.
3. The Bi-Level General Advantage Estimation breaks a long, single optimiation problem into a series of smaller, more manageable ones, making it possible to train the multi-turn reasoning VLMs.

## Weaknesses
1. The proposed visual reasoning framework is mainly tested on constrained, game-like environments (grid worlds, simple 3D navigation) with known coordinates. The success of Grounding and WorldModeling relies on the fact that the states in these tasks can be described with language. This may not hold in cluttered and dynamic environments with unpredictable elements.
2. Training efficiency could also be discusses, as the trajectory in this setting will be long and LLM-as-judge is used.
3. It is unclear that the method could be scaled up or transferred to other VLMs (other than Qwen-VL).

---

> ### Author Rebuttal · Authors · 2025-07-31
>
> We sincerely thank Reviewer 77zp for the excellent review and insightful feedback. We are greatly encouraged that the reviewer recognizes our work as complementary to existing research, praises our comprehensive experiments, and highlights the novelty and importance of the Bi-Level GAE method for enabling multi-turn visual reasoning.
>
> The reviewer’s main questions are related to the applicabitliy of our method to more complex environments, training efficiency, and its scalability to larger size and generalizability to other VLMs. We believe our clarifications and the new experimental results provided below will address these concerns.
>
> ---
>
> ### **1. (Regarding Weakness 1) Generalizability to Complex Environments**
>
> **TL;DR**: We chose controlled environments to enable fundamental and deep analysis. Our framework is directly applicable to most current agentic task simulators (e.g., AI2THOR, ALFWorld, WebShop, OSWorld) which provide text-based state information for supervision. The core principle of world modeling is essential for any agent's success, making our method broadly relevant.
>
> We thank the reviewer for this important point and acknowledge this limitation.
> 1.  **Rationale for Controlled Environments**: Our use of controlled environments was a deliberate choice. These settings are essential for conducting a rigorous and systematic analysis of the core components of visual reasoning. They allow us to isolate variables and understand the fundamental training dynamics.
> 2.  **Applicability to Current Agent Benchmarks**: We want to clarify that our framework is directly applicable to the majority of simulators used in current agentic research. Environments for embodied agents and GUI agents like **Maniskill, AI2THOR, ALFWorld, Minecraft, OSWorld, and WebShop** all provide access to text-based ground truth states that can be used as supervisory signals for grounding and worldmodeling. Since nearly all current agentic RL research is conducted in simulated environments, our method is highly relevant to the field's present state.
> 3.  **Fundamental Importance of Grounding and WorldModeling**: While understanding current state and predicting the next state might be challenging in cluttered and dynamic environments, an agent's ability to grounding and worldmodeling is fundamentally tied to its ability to understand the task and succeed. Therefore, our method is inherently correlated with task success and is broadly applicable in principle.
>
> ### **2. (Regarding Weakness 2) Discussion on Training Efficiency**
>
> **TL;DR**: We provide approximate training costs below and will add a detailed section on training efficiency to the appendix in camera ready.
>
> We appreciate the reviewer raising this practical concern. Training efficiency is indeed an important consideration. We have compiled the training costs for our primary experiments below.
>
> | Task | H100 GPU Hours (Approx.) | LLM-as-Judge Token Cost (Approx.) |
> | :--- | :---: | :---: |
> | FrozenLake | 40 | 2.3 x 10⁷ |
> | Sokoban | 40 | 2.3 x 10⁷ |
> | Navigation | 30 | 6.0 x 10⁶ |
> | ManiSkill | 30 | 4.6 x 10⁶ |
> | SVG | 10 | \ |
>
> We will add a dedicated section to the appendix of our camera-ready paper with a full analysis of these costs. We also note that efficiency could be further improved by integrating advanced techniques like async RL, which we are exploring as a future optimization.
>
> ### **3. (Regarding Weakness 3) Scalability and Transferability to Other VLMs**
>
> **TL;DR**: We ran new experiments on a larger model (Qwen2.5-VL 7B) and a different model family (InternVL3-2B). The results are highly positive, showing that our method can be scaled ups and generalizes across different VLM family.
>
> This is a crucial question, which we acknowledged as a limitation of our work in our submission draft. Now we are excited to share new results that address it directly. We conducted additional experiments on the Sokoban task using both a larger model from the same family (Qwen2.5-VL 7B) and a model from a different family (InternVL3-2B).
>
> | Model | Qwen2.5-VL 7B | InternVL3-2B  |
> | :--- | :---: | :---: |
> | **Base RL** | 0.625 | 0.360 |
> | **Visual Reasoning RL** | **0.922** | **0.389** |
>
> The results confirmed the scalibility and transferability of our method.

---

> > ### Comment · Reviewer_77zp · 2025-08-03
> >
> > I thank the authors for their response. The response has addressed most of my concerns. I'll maintain the score (5).

---

> ### Author Response · Authors · 2025-08-05
>
> Dear Reviewer 77zp,
>
> We sincerely thank you for recognizing the value of our work and for confirming that our rebuttal effectively addressed most of your concerns. The issues you raised regarding method applicability, training cost, scalability, and generalizability are all very important. We are pleased that our additional experiments and analyses helped clarify these points. We will incorporate these new results into the paper to ensure that your constructive feedback is fully reflected.
>
> Best,
> Authors of Submission 19716

---

### Official Review · Reviewer_HHiJ · 2025-07-03

**Clarity:** 4
**Significance:** 3
**Originality:** 3
**Rating:** 4
**Confidence:** 3

**Summary:**

This paper introduces Visual Reasoning Reinforcement Learning (VR-RL), aiming to strengthen visual state inference, and proposes the Bi-Level GAE method, which computes advantage estimates at both the turn level and the token level. Experimental results demonstrate that, compared to baseline RL methods, this approach significantly improves task performance and the quality of reasoning.

**Questions:**

Questions:
1. In Table 1, Grounding‐WorldModeling performs worse than WorldModeling on PrimitiveSkill. Could you provide some explanations and a failure analysis?
2. Table 1 presents the results of several “off-the-shelf VLMs.” In some tasks—particularly FrozenLake and SVG—these VLMs even outperform the Grounding-WorldModeling RL, despite having received no training. Does this suggest that the chosen tasks are too simple to showcase the advantages of RL, or that the success rate of this RL algorithm is not yet high enough?
3. The paper notes that structured representations, when inferred from "image-only input, are noisy due to limited grounding capabilities" in tasks like FrozenLake and Sokoban. Are there proposed architectural modifications or training paradigms that could improve the VLM's ability to autonomously generate accurate structured visual states from pixels, mitigating this performance degradation?
4. Visual Reasoning RL, incorporating the visual reasoning reward and Bi-Level GAE, leads to "more stable training and enhanced performance," with Base RL's performance sometimes declining in later stages while Visual Reasoning RL continues to increase. Could you explain what are the mechanistic reasons for Base RL's decline in performance during later training stages, and how does Bi-Level GAE specifically counteract this, leading to greater stability and sustained improvement?

**Ethical Concerns:**

["NO or VERY MINOR ethics concerns only"]

**Final Justification:**

The authors have provided clear and sufficient clarifications that address my initial concerns. In light of these responses, I find the work more convincing and thus raise my score to 4.

**Limitations:**

yes

**Quality:**

3

**Strengths And Weaknesses:**

Strengths:
1. The paper provides empirical evidence that incorporating explicit visual state reasoning into the VLM's thinking process enhances task performance.
2. The introduction of Visual Reasoning RL, which includes a turn-level visual reasoning reward and Bi-Level GAE, is a novel contribution that effectively addresses the complexities of credit assignment and reward propagation in multi-turn visual tasks.
3. The study systematically investigates and provides insights into the optimal choice of visual state representations, highlighting that natural language is generally effective, while structured formats are superior for manipulation-heavy tasks where precision is key and object-centric abstractions are available.

Weaknesses:
1. The paper notes a restricted model architecture as a limitation, suggesting that the findings might not generalize to all VLM families without further investigation.
2. The experimental scenarios are relatively simple, the set of evaluation metrics is limited, and the evaluation methodologies and choice of algorithms also exhibit certain inherent limitations.

---

> ### Author Rebuttal · Authors · 2025-07-31
>
> We sincerely thank Reviewer HHiJ for the detailed and constructive feedback. We are encouraged that the reviewer finds our **contributions novel** and **the empirical evidence compelling**.
>
> The reviewer's main concerns relate to the generalizability of our model architecture, the complexity of the experimental scenarios, and the need for deeper analysis on certain results and mechanisms. We believe our clarifications and the new experimental results provided below will address these points.
>
> ---
>
> ### **1. (Regarding Weakness 1) Generalizability of the Model Architecture**
>
> **TL;DR**: We ran new experiments on a different model family (InternVL3). The results show that our Visual Reasoning RL method generalizes and provides better performance compared Base RL.
>
> We appreciate the reviewer's point regarding the potential limitation of using a single model family. We chose Qwen2.5-VL because it is a powerful and widely used open-source VLM in RL research, allowing for a fair and controlled comparison with existing work. As the reviewer noted, we acknowledged this in the limitations section of the paper.
>
> To explicitly address this concern, we conducted new experiments on the **InternVL3-2B** model for the Sokoban task.
>
> | Method | **Test Success Rate** |
> | :--- |   :---: |
> | Base RL | 0.360|
> | **Visual Reasoning RL** | **0.389** |
>
> The results demonstratea that the Visual Reasoning RL generalizes to a different VLM and achieves higher performance compared to Base RL.
>
>
> ### **2. (Regarding Weakness 2) Simplicity of Scenarios, Limited Evaluation, and Choice of Algorithms**
>
> **TL;DR**: We argue that our tasks are challenging, as evidenced by the poor performance of existing powerful VLMs. While the reviewer’s comment that “the set of evaluation metrics is limited” and “the evaluation methodology and choice of algorithms exhibit inherent limitations” essentially reiterates points that we explicitly acknowledged in our limitations section, we agree this deserves clarification. We explain below why our current evaluation is sufficient and how our framework supports richer analysis. We also conducted new experiments with additional RL baselines to directly address the concern regarding algorithm selection.
>
> We thank the reviewer for encouraging us to clarify the complexity and scope of our evaluation.
>
> * **Task Complexity**: We respectfully disagree that our scenarios are simple. As shown in Table 1 of our paper, a highly capable model like **GPT-4o achieves an overall score of only 0.6**, which is far from the upper bound of 1.0. This indicates that the tasks are sufficiently challenging to differentiate even state-of-the-art models.
>
> * **Evaluation Methodology**: Regarding the concern on evaluation methodology, we want to emphasize that this is one of the points we explicitly listed in our limitations section. We focused on a small set of fundamental and widely recognized metrics in the main paper due to space constraints and to keep our analysis clear. Nevertheless, our evaluation is already extensive, covering **5 diverse tasks**, spanning **2D and 3D states**, and involving varied action spaces (**discrete, continuous, open-ended**), and evaluation metrics (**success rate, DINO score, DreamSim score**). Importantly, **our framework supports a wide range of metrics (e.g., format reward, grounding rewards, world-modeling rewards, entropy loss, response length, valid actions, effective actions, etc)**, which we will include in the appendix of the camera-ready version as auxiliary analyses.
>
> * **Choice of Algorithms**: Similarly, the point about the limited choice of algorithms is also one of the limitations we explicitly acknowledged. To further address this, we have now added experiments with additional RL baselines:
>
> | Method            | Overall Performance |
> | :---------------- | :-----------------: |
> | Vanilla-PPO       |         0.26        |
> | GRPO (w/mask)     |         0.54        |
> | Turn-PPO (w/mask) |         0.55        |
> | **Ours**          |       **0.82**      |
>
> *FrozenLake and Sokoban trained on 300 steps*
>
> These results demonstrate that our proposed algorithm significantly outperforms the additional RL baselines, further validating the effectiveness of our design choices.
>
>
> ### **3. (Regarding Question 1) Failure Analysis of Grounding-WorldModeling on PrimitiveSkill**
>
> This is an excellent question. In the PrimitiveSkill task, we give a list of object coordinates as the additional information. We hypothesize that the "Grounding" step, which involves translating these precise coordinates into a natural language description, can introduce noise. This linguistic abstraction is less useful when precise textual information is given. The agent can directly perform "WorldModeling" that leverages the structured input to predict the next state. This makes it inherently more robust for such precision-critical tasks. This insight is validated in Section 4 of our paper, where we show that structured representations are indeed better for PrimitiveSkill.
>
> ### **4. (Regarding Question 2) Off-the-Shelf VLMs Outperforming RL**
>
> We thank the reviewer for raising this important question about the performance comparison between off-the-shelf VLMs and our RL approach.
>
> 1.  **Task Complexity is Adequate**: As discussed previously (Regarding Weakness 2), we have explained that the tasks are challenging. Off-the-shelf VLMs do not solve them perfectly.
> 2.  **"No Training" is an Unverifiable Assumption**: The statement that off-the-shelf VLMs have received "no training" for the tasks we present here might not be accurate. Proprietary models like Claude and GPT are extensively fine-tuned, often with methods like RLHF, and their training data and processes are not public. They might also have been pre-trained or finetuned on the task we choose.
> 3.  **Our RL Methods on a 3B-Model Show Better Overall Performance**: Both Base RL and Visual Reasoning RL on a 3B-model outperform proprietary VLMs in overall scores. Proprietary VLMs used in our paper are often assumed to be significantly larger than 3B.
> 4.  **New Experiments with Extended Training Steps**: To further showcase the potential of our method, we extended the training on FrozenLake from 150 to 300 steps. The results show that our trained agent now surpasses the best-performing proprietary model evaluated for this task in the paper.
>
> | Model | FrozenLake Success Rate |
> | :--- | :---: |
> | Claude 3.7 Sonnet | 0.69 |
> | Qwen 2.5 VL 3B Untrained| 0.14 |
> | **Qwen 2.5 VL 3B Trained  with Visual Reasoning RL** | **0.71** |
>
> ### **5. (Regarding Question 3) Improving Generation of Structured Visual States**
>
> This is a very insightful question about a key challenge. We propose several training paradigms and architectural modifications to mitigate performance degradation from noisy grounding:
> 1.  **SFT**: We can first collect a dataset of scene images paired with their ground-truth text-based structured states. Fine-tuning the VLM (either the full model or just the vision tower) on this data would strengthen its ability to generate accurate structured representations for specific scenarios.
> 2.  **Integrating Better Vision Encoder**: We can incorporate existing fine-grained image understanding approaches like GLIP, RegionCLIP, and FG-CLIP into the VLM architecture. These models are explicitly trained for enhanced grounding and could provide more robust object-centric features.
> 3.  **Scaffolding and Prompting**: Techniques that guide the VLM's internal processing, such as prompting it to generate a "cognitive map" or spatial representation of the scene before outputting the final structured state, could improve accuracy.
>
> References:
>  *Li et al., "Grounded Language-Image Pre-training", CVPR 2022.*
>  *Zhong et al., "RegionCLIP: Region-based Language-Image Pretraining", arXiv 2021.*
>  *Xie et al., "FG-CLIP: Fine-Grained Visual and Textual Alignment", arXiv 2025.*
>  *Yin et al., "Spatial Mental Modeling from Limited Views", arXiv 2025.*
>
> ### **6. (Regarding Question 4) Stability of Bi-Level GAE vs. Base RL's Decline**
>
> This question gets to the core of our method's advantage. The performance decline in Base RL during later training stages is a problem in sparse-reward, multi-step tasks.
> *   **Why Base RL Fails**: In later training stages, the agent has already mastered the "easy" trajectories. For more challenging cases that require precise, long-horizon reasoning, Base RL provides only a single reward at the very end. If the agent executes several correct turns but makes one mistake at the final step, the entire sequence of correct actions is penalized along with the final error because the trajectory-level advantage is negative. This noisy credit assignment discourages the necessary exploration to solve hard cases and leads to training instability.
> *   **How Bi-Level GAE Provides Stability**: Our Bi-Level GAE directly counteracts this. By computing advantages at both the trajectory level and the turn/token level, it provides **turn-level rewards**. Even if a trajectory fails at the end, our method can assign positive rewards to the intermediate turns that were executed correctly. This fine-grained feedback allows the agent to learn progressively, reinforcing correct sub-policies even when tackling difficult problems.

---

### Author Response · Authors · 2025-08-09
**Summary of Responses to Reviewers**

We thank the AC and all reviewers for their constructive feedback and engagement. Multiple reviewers recognized the novelty of integrating **Visual Reasoning RL** with a **Bi-Level GAE** mechanism for multi-turn VLM agents, as well as the breadth of our evaluation. We summarize our thesis, key results, and how we addressed core concerns below.

**Thesis**

Current VLMs often struggle in partially observable, multi-turn environments due to coarse credit assignment and insufficient visual state reasoning. We introduce *Visual Reasoning RL*, which incorporates explicit visual reasoning rewards and Bi-Level GAE to provide fine-grained credit assignment at both trajectory and turn levels. Across five diverse environments, our method substantially improves task success rates and reasoning quality compared to base RL methods.

**Key results**

* **Reasoning + fine-grained credit assignment wins:** Visual Reasoning RL improves overall performance by **+49%** over the next-best RL baseline.
* **Generalization:** Gains hold across different VLM families (Qwen2.5-VL, InternVL3) and scales to larger models (3B -> 7B).
* **Beating strong proprietary models:** Our 3B model surpasses GPT-o3 and GPT-o4-mini on most tasks despite using fewer parameters and smaller context budgets.
* **Sample efficiency:** Visual Reasoning RL accelerates early learning in most tasks.
* **Robustness:** Bi-Level GAE mitigates late-stage performance decline in base RL via fine-grained credit assignment to identify correct intermediate steps.

**Rebuttal summary**

1. **Model architecture generalizability (Reviewer HHiJ, Reviewer 77zp)**

   * Added experiments with InternVL3-2B and Qwen2.5-VL 7B, confirming consistent gains beyond the original model family.

2. **Task complexity & evaluation methodology (Reviewer HHiJ, Reviewer ortS)**

   * Clarified tasks’ adjustable difficulty and challenging nature (e.g., GPT-4o scores 0.6, which is far from upper-bound 1.0).
   * Expanded metric coverage (success rate, DINO, DreamSim, format reward, grounding/world-modeling rewards, entropy, valid actions) for camera-ready.

3. **Choice of algorithms (Reviewer HHiJ, Reviewer ortS)**

   * Added baselines: Vanilla-PPO, GRPO (w/mask), Turn-PPO (w/mask); Visual Reasoning RL leads next-best by +49%.

4. **PrimitiveSkill performance (Reviewer ortS)**

   * Clarified that gains are in early sample efficiency; later convergence is due to reduced grounding demands.

5. **Stronger baselines & scalability (Reviewer 77zp, Reviewer ortS)**

   * Benchmarked against GPT-o3, GPT-o4-mini; our 3B model achieves higher scores on most tasks.

6. **Bi-Level GAE broader impact (Reviewer wZ7m)**

   * Tested on FreeThink strategy; confirmed improvements, especially with denser intermediate rewards.

7. **SVG task inclusion (Reviewer wZ7m)**

   * Motivated as an open-ended, multi-turn reformulation with diverse evaluation metrics, informed by prior multi-turn reasoning literature.

8. **Variance & stability (Reviewer ortS)**

   * Reported mean ± std over multiple seeds for key tasks; provided extended training curves showing stability benefits.

All clarifications, expanded analyses, and new results will be incorporated into the final version.

---

### Decision · Program_Chairs · 2025-09-17

**Decision:**

Accept (poster)

**Comment:**

The authors explore improving the performance of vision-language models in multi-turn agentic tasks in interactive dynamic environments (modeled as POMDPs), examining the effect of methods enabling reasoning about the visual states and world models. Regarding reasoning about visual states, they compare multiple reasoning strategies (No-Think, Free-Think, Grounding, WorldModeling, Grounding-WorldModeling). Regarding visual state representations, the authors consider several variants (natural language descriptions, structured formats, symbolic representations). Regarding RL-based optimization over a sequence of turns, the authors propose Bi-Level General Advantage Estimation using turn-level visual reasoning rewards with a LLM-as-a-Judge (LLMaaJ) and the token level for action generation. This combination of methods consistently (but not universally) outperform multiple baseline foundational models (VLM-R1-3B, Qwen2.5-VL{3,7,72}B, InternVL3-2B, GPT-4o, Gemini 2.0, Claude 3.7 Sonnet) over challenging benchmark environments (Sokoban, FrozenLake, Navigation, PrimitiveSkill, SVG Reconstruction). Secondary experiments are performed to validate the individual component strategies and point towards directions for future work.

Strengths identified by the reviewers include:
- Multi-turn RL-trained VLM reasoning models haven't been studied in detail, are important (and likely more so in the future), and the authors address several issue in accruing improvements.
- Strong evidence is problem that incorporating visual state reasoning, especially WorldKnowledge and WorldKnowledge+Grounding, into the VLM reasoning trajectory improves task performance.
- The Bi-Level GAE approach is novel and enables efficient RL learning.
- The additional experiments regarding visual state representation is insightful and provides task-family specific guidance that also aligns with intuition. The empirical evaluation is relatively extensive overall.
- The paper is well-written and easy to understand while the formulation and considered alternatives are well-motivated.

Weaknesses identified by the reviewers include:
- Some configurations require additional interpretation (e.g., Grounding-WorldKnowledge vs. WorldKnowledge (reviewer HHiJ), off-the-shelf model performance (reviewers HHiJ, 77zp), Bi-Level GAE stability explanation (reviewer HHiJ), PrimitiveSkill task concerns (reviewer ortS)). However, these were well-rebutted during the discussion period.
- The experiments are conducted primarily on controlled environments, which partially explain why structured representations and WorldKnowledge states are helpful for improving performance. However, this is likely to transfer to a large set of practical problems also.
- The empirical results are consistently better than baseline models being augmented, but fall short of proprietary models in some cases -- meaning there is a better method out there (even if it may require a lot more data and training). This was addressed in rebuttal with longer training runs, but would still need to be integrated into the paper.
- [my own criticism] The paper reads a bit like a set of explorations instead of a coherent formulation -- many which have analogues in text-based multi-turn RL. One could imaging a 'truly exhaustive' set of ablations where you could tease apart the individual contributions and secondary experiments that emphasize interpretation of the differences in performance. The current version reads a bit like a disjoint set of findings (modulo the Bi-Level GAE formulation). Some of this can be addressed in preparing a revised version (and maybe a bit of restructuring the paper would fit better with how I interpreted the contributions), but some would require additional experiments and interpretation.

Overall, the Bi-Level GAE is innovative for multi-turn RL with VLM agents and the findings regarding visual state type and representation are insightful. Additionally, the performance consistently shows improvement over 'vanilla' foundational models. In my reading of the paper, I would estimate it to be borderline (and leaning toward reject), but with the improved results relative to proprietary models and rebuttal overall, I lean more towards accepting as the findings are promising and I expect will be cited as a baseline in future works in this area of growing interest.